

# Tide characteristics and tidal wave propagation in the Persian Gulf

S. Mahya Hoseini[1], Mohsen Soltanpour[2]

[1]Civil Engineering Department, K. N. Toosi University of Technology, Tehran, P.C. 19967-15433, Iran
[2]Civil Engineering Department, K. N. Toosi University of Technology, Tehran, P.C. 19967-15433, Iran

*Correspondence to*: S. Mahya Hoseini (sm.hoseini@mail.kntu.ac.ir)

**Abstract.** A 2D hydrodynamic model is employed to study the characteristics of tidal wave propagation in the Persian Gulf (PG). The study indicates that tidal waves propagate from the Arabian Sea and the Gulf of Oman into the PG through the Strait of Hormuz. The numerical model is first validated using the measured water levels and current speeds around the PG and the

principal tidal constituents of Admiralty tide tables. Considering the intermediate width of the PG, in comparison to Rossby deformation radius, the tidal wave propagates like a Kelvin wave on the boundaries. Whereas the continental shelf oscillation resonance of the basin is close to the period of diurnal constituents, the results show that the tide is mixed mainly semidiurnal. A series of numerical tests is also developed to study the various effects of geometry and bathymetry of the PG, Coriolis force, and bed friction on tidal wave deformation. Numerical tests reveal that the Coriolis force, combined with the geometry of the

gulf, results in generation of different amphidromic systems of diurnal and semidiurnal constituents. The configuration of the bathymetry of the PG, with a shallow zone at the closed end of the basin that extends along its longitudinal axis in the southern half (asymmetrical cross section), results in the deformations of incoming and returning tidal Kelvin waves and consequently the shifts of amphidromic points (APs). The bed friction also results in the movements of the APs from the centerline to the south border of the gulf.

**1 Introduction**

Understanding the dynamics of tides is essential to improve the performance of systems dealing with water level fluctuations, e.g., navigation, coastal engineering projects, fishing, and generation of tidal energy. Tide-generating forces produced by gravitational interaction between the moon, the sun, and the earth, result in the generation of the astronomical tides. Tides can be described as the sum of two types of oscillations: (1) the co-oscillating tide, caused by tidal waves penetrating from the

adjacent ocean or sea. (2) the independent tide, generated directly by the tide generating forces that prevails where tidal waves from adjacent basins cannot significantly penetrate the sea (Defant, 1961; Medvedev et al., 2019).

Tides have very different dynamics in shallow water depths compared to deep waters. Tidal propagation and ranges in shallow waters of the continental shelves, gulfs, straits, and estuaries are severely affected by various factors such as the basin geometry, friction, and Coriolis. The tidal range is increased by shoaling in a landward direction due to the gradual decrease of water

depth. Funneling also results in the increase of the tidal wave height in converging channels. In narrow gulfs, when the





propagating tide meets the tide reflecting from the closed end of the gulf, a standing wave can be formed. Resonance occurs when the length of the gulf is about one-quarter of the standing wavelength (or odd multiples of it), which results in a high tidal range at the head of the gulf. Tidal wave heights are also damped due to bottom friction.

The Coriolis force deflects tidal waves across the gulf to the right in the northern hemisphere and to the left in the southern
hemisphere (Allen, 2009; Van Rijn, 2010). An amphidromic system develops as the result of the combined constraint of Coriolis force effect and ocean basin geometry. During each tidal period, the tidal wave crest circulates around an Amphidromic Point (AP) at high water. The tidal range is zero at each AP and increases away from it. Co-tidal lines connect the points where the tide is at the same phase of its cycle, and co-range lines link the points where the tidal range is equal (Wright et al., 1999).

Tides in the Persian Gulf (PG) co-oscillate with those in the narrow Strait of Hormuz, which opens into the deep Gulf of Oman, where the tides co-oscillate with those in the Arabian Sea. The dimensions of the PG are such that resonance amplification of the tides can occur. This results in two APs of semidiurnal constituents in the northwest and southeast ends and a single AP of diurnal constituents in the center near Bahrain (Reynolds, 1993).

Krummel (1911) defined the tide in the PG as a progressive wave that penetrates through the Strait of Hormuz, clinging to the
land on the right side and traveling counter-clockwise around the whole PG. Defant (1961) also stated that the tide in the PG is a standing wave in which the rotation of the earth deflects it and converts the nodal lines of different constituents to APs. Reynolds (1993) interpreted the PG tides as complicated standing waves with the primarily semidiurnal and diurnal patterns at different locations.

Literature shows a number of numerical studies of the tidal wave propagation in the PG. Elahi and Ashrafi developed a two-
dimensional numerical model to analyze the dynamics of the four major tidal constituents $M_2$, $S_2$, $K_1$, and $O_1$ in the PG (Elahi and Ashrafi, 1994). They also presented the co-tidal/co-range charts and velocity fields of major tidal constituents. Najafi et al. (1997) modelled the tides of the PG and presented the contour charts of tidal amplitudes and phases for the major tidal constituents, tidal current ellipses, depth-averaged velocities, and residual tidal currents. Kämpf and Sadrinasab, (2006) employed a three-dimensional hydrodynamic model (COHERENS) to study the circulation and water mass properties of the
Persian Gulf. Sabbagh-Yazdi et al. (2007) applied the water level data of Admiralty tide tables (UKHO, 2005) at the open boundary of Hormuz Strait in their model to extract the co-tidal charts of the PG. Pous et al. (2013) applied a 2D shallow-water model over the northwestern Indian Ocean, forced by seven tidal components at the southern boundary, to derive the co-tidal/co-range charts of harmonic constituents of the PG. They also presented velocities of tidal currents, residual tidal currents, and form factor over the PG. Akbari et al. (2016) applied the finite volume model of FVCOM (Chen et al., 2003), forced by
eight tidal constituents at its southern boundary to study tidal amplitudes in an extended domain comprising of the PG, Gulf of Oman, and the Arabian Sea and presented the co-tidal/co-range charts. Implementing the 2D shallow-water model of VOM-SW2d (Backhaus, 2008), Mashayekh Poul (2016) presented the co-tidal/co-range charts, maximum velocities, tidal ellipses, and kinetic power energy for the PG using 13 tidal constituents at the open boundary in the Gulf of Oman. They also studied the resonance characteristics of the PG tide by a numerical model, ascertained with 33 runs with different periods. Using the





two-dimensional hydrodynamic model of MIKE21 software, Sohrabi Athar et al. (2019) presented a tidal model for the PG with spatially variable bed friction coefficients, forced by satellite altimetry sea level data. Ranji and Soltanpour (2021) showed that the spatially varying friction, as a function of water depth, mean velocity, vegetation, and bed sediment size, results in a slight overall improvement of model outputs, compared to applying constant friction in the Persian Gulf.

Despite all past efforts, tidal modeling in the PG still needs to be improved, with comparisons to new water levels and current
speeds measurements at different locations. Moreover, unlike similar studies in other basins such as the Yellow Sea, China Sea, and Gulf of Thailand (Su et al., 2015; Zu et al., 2008; Phan et al., 2019; Tomkratoke et al., 2015; Cui et al., 2019), here the analysis of existing data and the behavior of the influencing factors on tidal wave dynamics are limited.

The present study aims to set up a comprehensive and precise tidal model in the PG to figure out the dominant physical mechanisms required to reproduce the main features of the observed patterns of tide propagation in the PG. The high-resolution
bathymetry of the PG is introduced to the 2D hydrodynamic model, and the calibrations, as well as sensitivity analysis of model inputs, are conducted by numerous measured water levels and tidal constituents around the PG. Different charts of co-tidal, co-range, shallow-water constituents, maximum tidal range, shape factor, and effective resonance period are extracted from the model results. The formation of the amphidromic system in the PG and the effects of important governing factors on tidal behavior, i.e., the Coriolis force, friction, and bathymetry, are also studied.

**2 Study area and field measurements**

**2.1 The Persian Gulf**

The shallow semi-enclosed basin of the PG locates in the south of Iran between latitudes 24°-30°N and longitudes 48 °-56 °E (Fig. 1). The PG is connected to the Arabian Sea through the Gulf of Oman by the narrow Strait of Hormuz. Its average depth is about 36 m with the maximum width and length of about 338 km and 1000 km, respectively. The tide in the PG is very
complicated, and the dominant pattern, i.e., primarily semidiurnal or diurnal, varies from one region to another (Reynolds, 1993).

The PG's natural period of waves is calculated as 22.6 and 21.7 hours based on the Japanese and Chrystal methods, respectively (Defant, 1961).





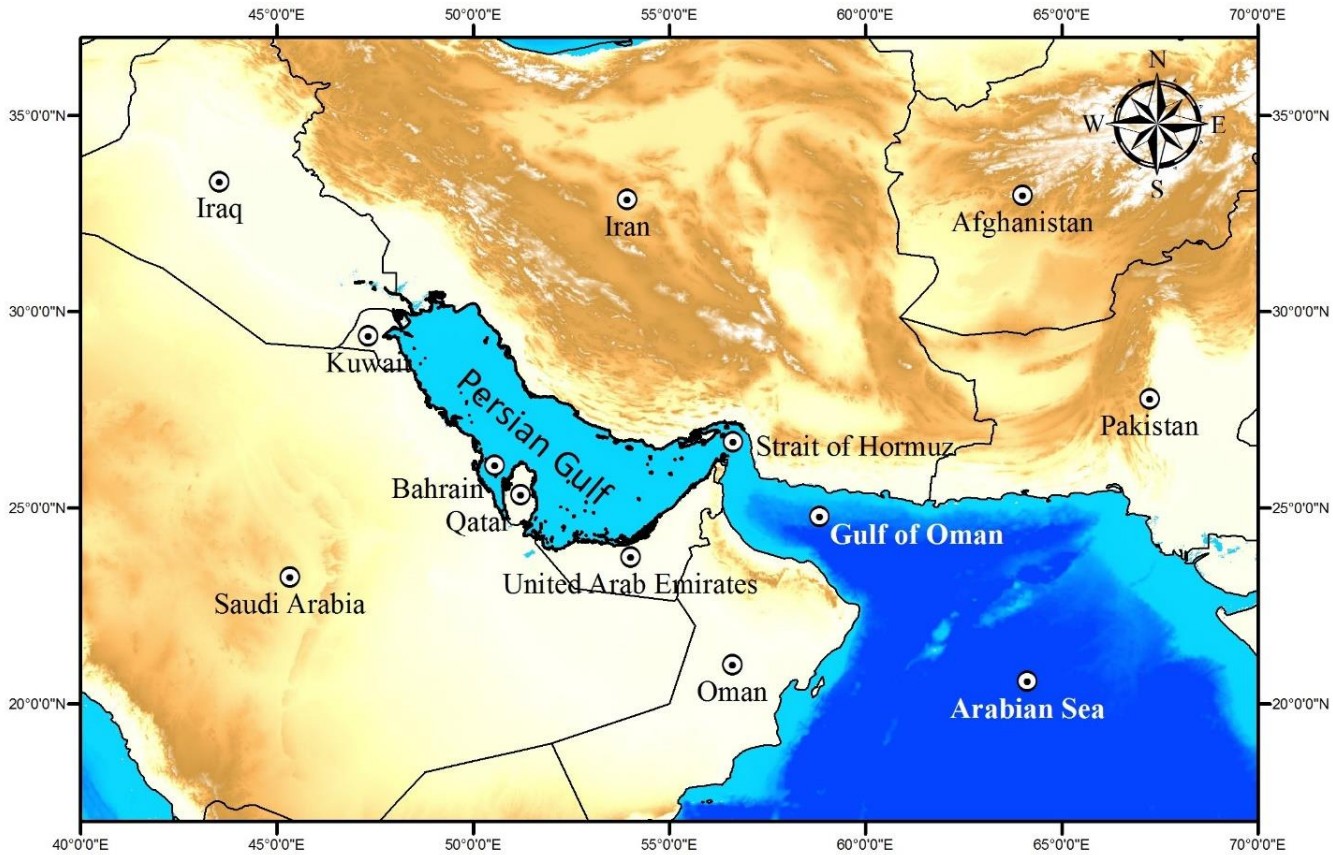

**Figure 1: Geography of the Persian Gulf (NOAA 2006).**

**2.2 Field measurements**

The field measurements are mostly selected from the monitoring and modeling studies in the northern coastline of the PG
(PMO, 2015). Tide Gauges (TGs) and AWACs (Acoustic Wave and Current), which are Nortek ADCPs (Acoustic Doppler
Current Profiler), were employed in these studies. Taking the advantage of the simple echo sounder principle, a vertically
oriented transducer in the center of the AWAC was used to directly measure the distance to the sea surface. Table 1 lists the
properties of the aforementioned field measurements. Limited water level data are also available on the south coastline of the
PG (Elhakeem et al., 2015; Pokavanich et al., 2015; Rakha et al., 2008; NOAA, 2015). The locations of all water level
measurements are illustrated in Fig. 2. In addition to water level measurement data, two large sets of tidal harmonic
constituents, derived from water level observations, are also employed (Fig. 3). The first set is provided by the National
Cartographic Center of Iran (NCC) using the Institute of Ocean Sciences (IOS) method in the north of the PG (NCC, 2016).
The second one is provided by U.K Hydrographic Office around the coastlines of the PG (UKHO, 2005) using the Admiralty
method (Glen, 2015).



**Table 1: Field measurements (PMO, 2015).**

| Station | Instrument (Depth, m) | Measurement Period | | Station | Instrument (Depth, m) | Measurement Period | |
|---|---|---|---|---|---|---|---|
| | | Start | End | | | Start | End |
| Deylam | AWAC (15) | 2010/07/09 | 2010/07/09 | Kish Island | TG (3.5) | 2010/01/02 | 2010/04/14 |
| Deylam | TG (3) | 2010/07/11 | 2011/08/11 | Farur Island | AWAC (25) | 2009/09/10 | 2010/10/12 |
| Genaveh | TG (3) | 2010/06/25 | 2011/08/20 | Bustaneh | TG (4) | 2009/11/25 | 2010/08/18 |
| Kharg | AWAC (25) | 2010/ 07/31 | 2011/07/29 | Basaidu | TG (5) | 2009/10/22 | 2009/12/25 |
| Bandar Bushehr | TG (3) | 2010/06/24 | 2011/08/07 | Khamir | TG (4.5) | 2009/08/07 | 2009/09/11 |
| Lavar | TG (3) | 2010/06/24 | 2011/08/19 | Pohl | TG (5.8) | 2009/08/07 | 2010/08/15 |
| Dayyer | TG (3) | 2008/08/23 | 2009/09/21 | Kaveh | AWAC (10) | 2013/06/16 | 2013/06/30 |
| Kangan | AWAC (22) | 2008/08/23 | 2009/09/25 | Dargahan | TG (7) | 2009/08/08 | 2009/09/15 |
| Taheri | TG (3) | 2008/08/23 | 2009/09/21 | Shahid Rajaee | TG (5.2) | 2009/08/08 | 2010/02/01 |
| Nayband | TG (3) | 2008/08/23 | 2009/09/21 | Bahman | TG (3.5) | 2009/10/21 | 2009/12/28 |
| Lavan Island | AWAC (25) | 2009/08/12 | 2010/07/08 | Larak Island | TG (3) | 2009/08/09 | 2010/08/14 |
| Chiruyeh | TG (5) | 2009/08/06 | 2010/08/18 | Sirik | TG (5.8) | 2009/08/10 | 2010/09/04 |


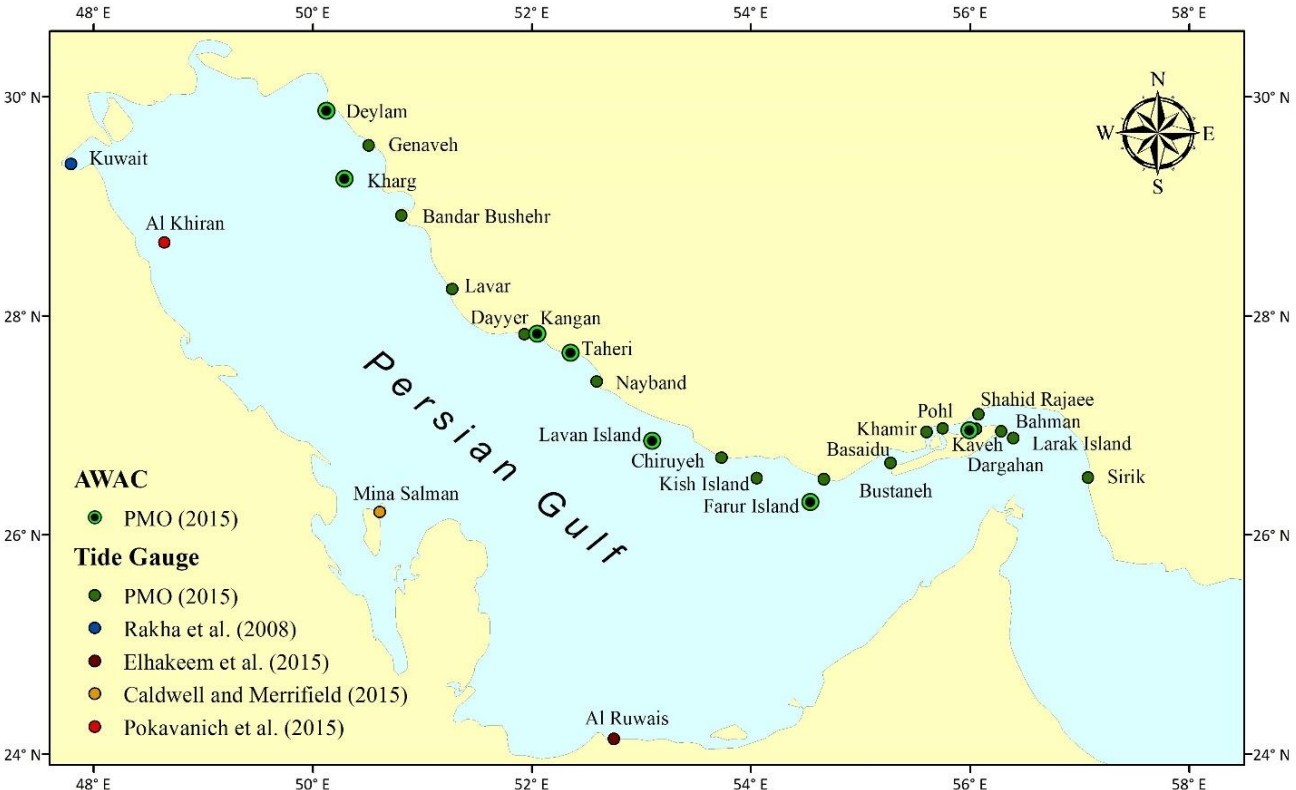

**Figure 2: Water level measurements around the PG.**



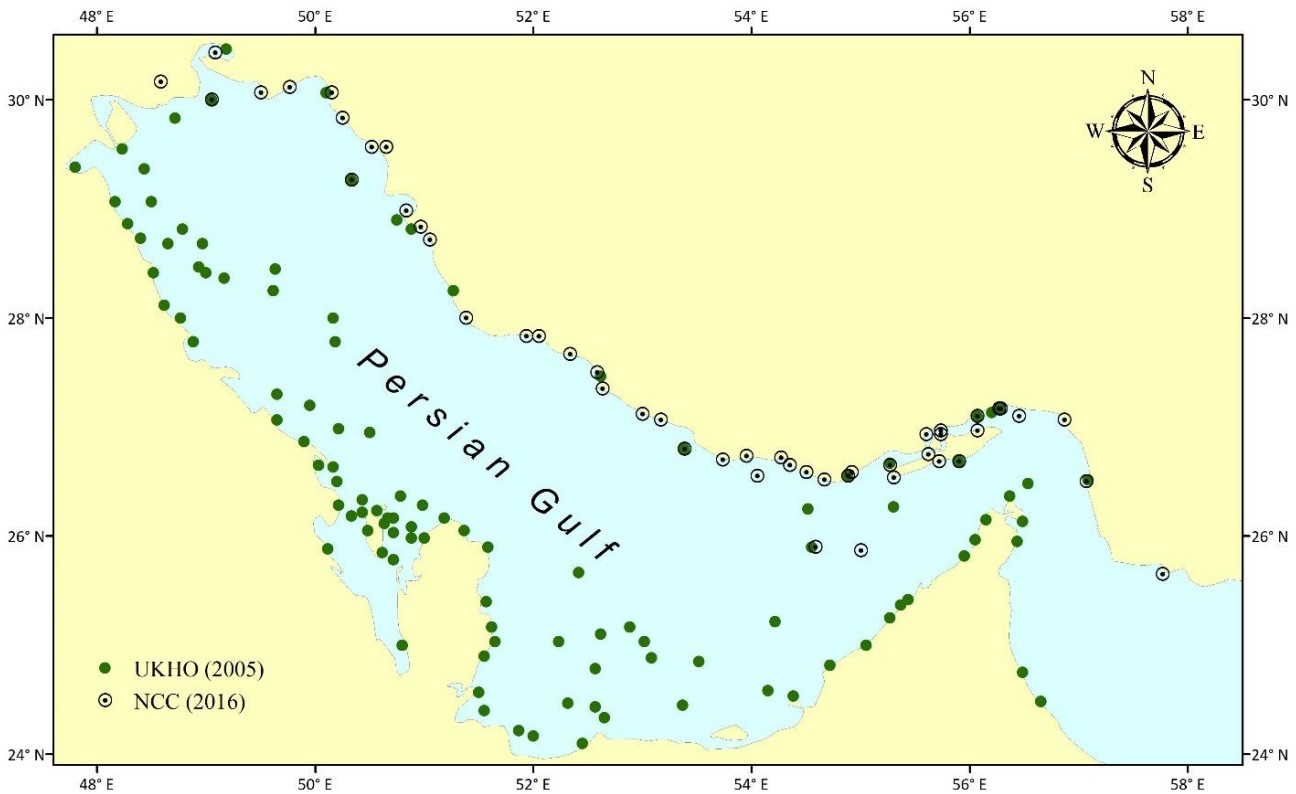

**Figure 3: Harmonic constituent data of NCC and UKHO.**

Excluding the stations with relatively short duration of measurements, i.e., less than six months, major tidal constituents of the northern coastline of the PG are extracted by harmonic analysis (Foreman et al., 2009). Following the Rayleigh criterion on the shortest length of observational sea level time series, 60 tidal constituents can be extracted by harmonic analysis (Foreman, 1979). Five categories are introduced to classify the constituents in about the one-year observed tide (Table 2). Shallow-water constituents depend on the depth and long-period, diurnal and semidiurnal categories are defined based on the period of

constituent. Although the origin of $M_3$ is lunar, its period is equal to 8.2 hours and hence it is included in a different category (Parker, 2007).

**Table 2: Categories of tidal constituents.**

| Category | Constituent |
|---|---|
| Long-period | MM, MSF, MF, SSA,MSM |
| Diurnal | $2Q_1$, $Q_1$, RHO, $O_1$, $P_1$, $K_1$, $J_1$, $OO_1$,$TAU_1$, $BET_1$, $SO_1$ , $NO_1$ , $PHI_1$, $THE_1$ , $CHI_1$ , $UPS1$ , $SIG_1$ , $ALP_1$ |
| Semidiurnal | $2N_2$, $MU_2$, $N_2$, $NU_2$, $M_2$, $L_2$, $S_2$, $K_2$, $MSN_2$, $LDA_2$, $ETA_2$, $OQ_2$ , $MKS_2$ , $EPS_2$ |
| Shallow-water | $MK_3$, $MN_4$, $M_4$, $MS_4$ , $S_4$, $MO_3$, $SO_3$,$SK_3$, $2MK_5$, $2MK_6$ , $2MS_6$ , $M_6$, $SK_4$, $SN_4$ , $MSK_6$ , $2MN_6$ , $2SK_5$, $MK_4$, $3MK_7$, $2SM_6$ , $M_8$ |
| Other | $M_3$ |





Figure 4 shows the categories of tidal constituents on the north coastline of the PG. It is observed that in general the largest

contributions are semidiurnal, followed by diurnal, long-term, and shallow-water constituents, respectively.

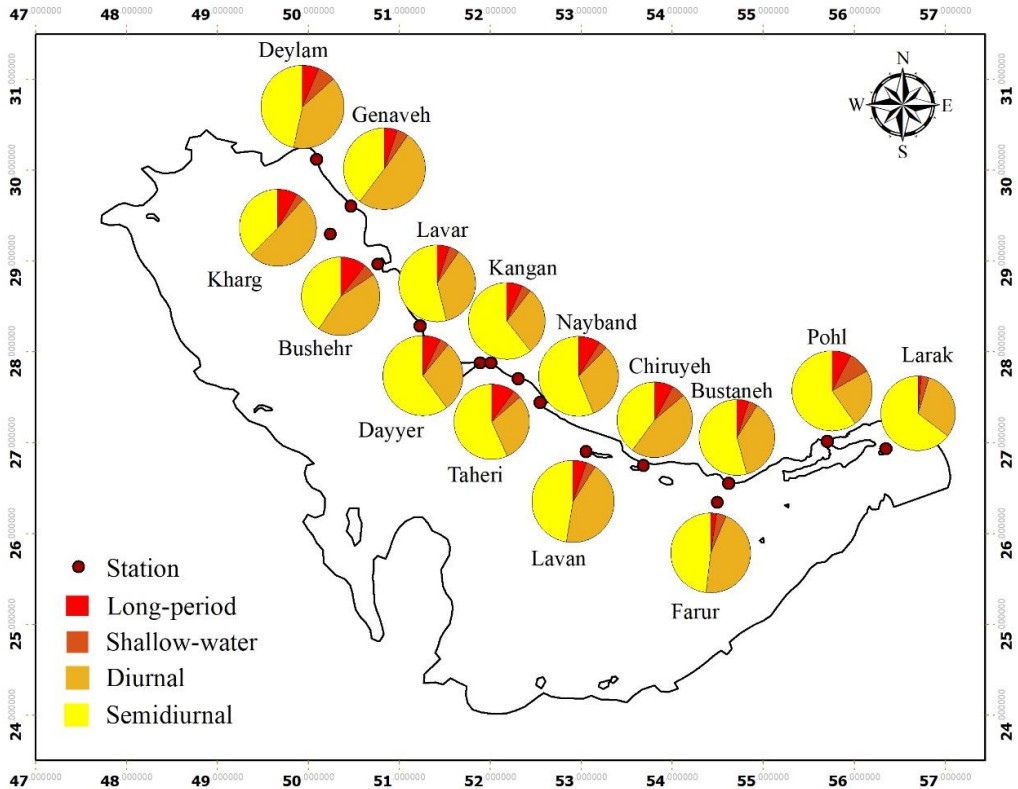

**Figure 4: Contribution of tidal constituents on the northern coastline of the PG.**

## 3 Numerical model

### 3.1 Model set up

#### 3.1.1 Bathymetry and computational grid

Figure 5 shows the sources of bathymetry data, i.e., 1:25k and 1:100k hydrographic maps along the northern coastline of the PG by NCC and 2-minute gridded global relief data (ETOPO2 v2) in other areas (NGDC, 2006). The reference of NCC data is shifted from Chart Datum (CD) to MSL using the difference level at each hydrography map. The shoreline is extracted from high-resolution dataset of GSHHG (Wessel, Pål, 1996). Figure 6 presents the final bathymetry of the model. The mangrove

forests and salt marshes at the Khuran Channel, located north of the Qeshm Island, are excluded from the computational domain to improve the modeling results (Fig. 7).





The numerical modeling is conducted using the Flow Model (FM) module of MIKE 21 (DHI, 2012), based on the numerical solution of the 2D shallow-water equations resulting from the depth-integrated incompressible Reynolds averaged Navier-Stokes equations. Figure 6 shows the domain of the numerical model from 47.6°-58° E and 23.8°-30.4° N. The unstructured

mesh comprises 16664 nodes and 31239 elements with a grid cell size of about 163 km$^2$ that gradually decreases to $4.1 \times 10^4$ m$^2$ in shallow areas. The number of grid points is also increased near mangrove locations in the Khuran Channel to improve the modeling results (Fig. 7). Sensitivity analysis reveals that the grid is fine, and there is no need to increase the nodes and elements.

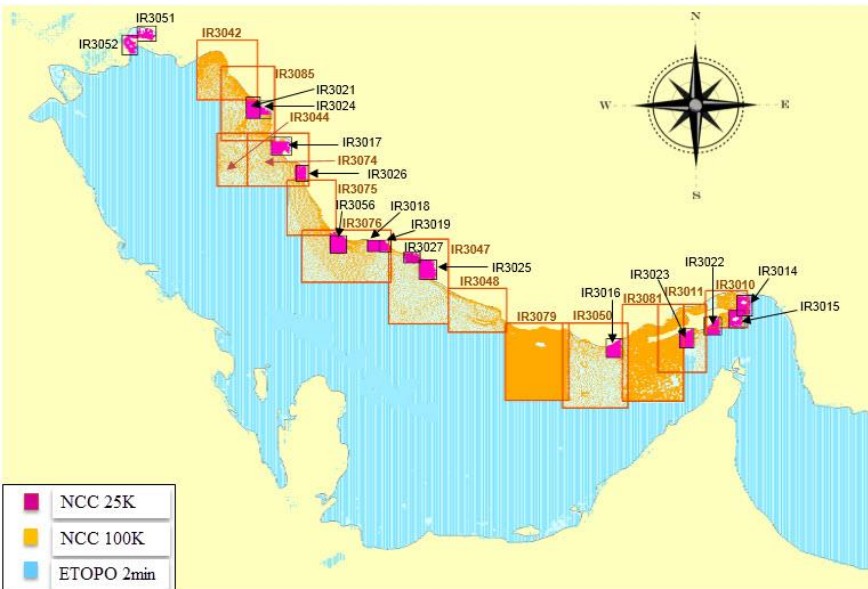

**Figure 5: Bathymetry data sources from ETOPO and NCC.**



**Figure 6: Bathymetry and computational grid.**





### 3.1.2 Bed friction

Table 3 lists the statistical comparison of 1-hour modeled and observed water levels for Manning numbers of 50, 60, and 70 $m^{1/3}$ $s^{-1}$ in all stations. The optimum value of Manning roughness coefficient can be selected by statistical analysis, i.e. comparing the Correlation, Coefficient of determination, and Root Mean Square Error (RMSE) indices:

$$Correlation = \frac{\sum_{i=1}^{k}(A_m - \overline{A_m})(A_o - \overline{A_o})}{\sqrt{\sum_{i=1}^{k}(A_m - \overline{A_m})^2 \ \sum_{i=1}^{n}(A_o - \overline{A_o})^2}} \qquad (1)$$

$$Coefficient\ of\ determination = 1 - \frac{\sum_{i=1}^{k}(A_m - A_o)^2}{\sum_{i=1}^{k}(A_o - \overline{A_o})^2} \qquad (2)$$

$$RMSE = \sqrt{\frac{\sum_{i=1}^{k}(A_m - A_o)^2}{k}} \qquad (3)$$

where $A_m$, $A_o$, and $k$ denote the modeled water level, observed water level (or amplitude of tidal constituent), and the number
of samples, respectively. Although the differences of statistical measures are not significant, it is observed that all three indices correspond to the optimum Manning number of 60 $m^{1/3}$ $s^{-1}$.

**Table 3: Statistical comparison of modeled and observed water levels for different Manning numbers.**

| Station | RMSE | | | Coefficient of determination | | | Correlation | | |
|---|---|---|---|---|---|---|---|---|---|
| | $n=50$ | $n=60$ | $n=70$ | $n=50$ | $n=60$ | $n=70$ | $n=50$ | $n=60$ | $n=70$ |
| Bustaneh | 0.14 | 0.1 | 0.1 | 0.92 | 0.96 | 0.95 | 0.96 | 0.98 | 0.98 |
| Chiruyeh | 0.13 | 0.15 | 0.15 | 0.89 | 0.85 | 0.83 | 0.94 | 0.94 | 0.94 |
| Shahid Rajaee | 0.17 | 0.16 | 0.17 | 0.96 | 0.96 | 0.96 | 0.98 | 0.98 | 0.98 |
| Khamir | 0.28 | 0.25 | 0.22 | 0.92 | 0.94 | 0.95 | 0.98 | 0.98 | 0.98 |
| Pohl | 0.32 | 0.29 | 0.34 | 0.88 | 0.91 | 0.88 | 0.95 | 0.97 | 0.96 |
| Larak | 0.24 | 0.15 | 0.15 | 0.88 | 0.95 | 0.96 | 0.94 | 0.98 | 0.98 |
| Taheri | 0.19 | 0.15 | 0.14 | 0.83 | 0.88 | 0.89 | 0.91 | 0.94 | 0.94 |
| Nayband | 0.18 | 0.15 | 0.14 | 0.83 | 0.87 | 0.88 | 0.91 | 0.94 | 0.94 |
| Dayyer | 0.25 | 0.17 | 0.16 | 0.75 | 0.88 | 0.89 | 0.87 | 0.94 | 0.94 |
| Dargahan | 0.36 | 0.34 | 0.35 | 0.85 | 0.87 | 0.84 | 0.92 | 0.93 | 0.92 |
| Bushehr | 0.23 | 0.25 | 0.17 | 0.72 | 0.67 | 0.84 | 0.92 | 0.93 | 0.96 |
| Genaveh | 0.12 | 0.11 | 0.11 | 0.94 | 0.95 | 0.95 | 0.98 | 0.98 | 0.98 |
| Deylam | 0.16 | 0.14 | 0.13 | 0.93 | 0.95 | 0.95 | 0.98 | 0.98 | 0.98 |
| Kharg | 0.1 | 0.07 | 0.1 | 0.94 | 0.97 | 0.94 | 0.98 | 0.99 | 0.97 |
| Lavar | 0.11 | 0.09 | 0.09 | 0.93 | 0.95 | 0.95 | 0.99 | 0.98 | 0.98 |
| Lavan | 0.08 | 0.07 | 0.11 | 0.96 | 0.96 | 0.90 | 0.98 | 0.98 | 0.98 |
| Farur | 0.11 | 0.1 | 0.12 | 0.94 | 0.94 | 0.90 | 0.97 | 0.97 | 0.98 |
| Sirik | 0.27 | 0.26 | 0.31 | 0.85 | 0.86 | 0.71 | 0.93 | 0.93 | 0.90 |
| Bahman | 0.29 | 0.26 | 0.30 | 0.85 | 0.88 | 0.83 | 0.92 | 0.94 | 0.90 |
| Basaidu | 0.19 | 0.26 | 0.19 | 0.92 | 0.87 | 0.92 | 0.97 | 0.93 | 0.96 |
| Kish Island | 0.16 | 0.17 | 0.16 | 0.84 | 0.84 | 0.87 | 0.92 | 0.91 | 0.93 |
| Kaveh | 0.34 | 0.37 | 0.37 | 0.88 | 0.88 | 0.87 | 0.95 | 0.94 | 0.93 |
| Al Ruwais | 0.24 | 0.26 | 0.28 | 0.72 | 0.68 | 0.61 | 0.85 | 0.85 | 0.84 |





| | | | | | | | | | |
|---|---|---|---|---|---|---|---|---|---|
| Mina Salman | 0.23 | 0.22 | 0.21 | 0.78 | 0.81 | 0.82 | 0.94 | 0.94 | 0.93 |
| Al Khiran | 0.25 | 0.23 | 0.27 | 0.79 | 0.83 | 0.76 | 0.93 | 0.92 | 0.88 |
| Kuwait | 0.52 | 0.48 | 0.46 | 0.77 | 0.81 | 0.83 | 0.97 | 0.96 | 0.95 |
| **Average** | **0.217** | **0.201** | **0.204** | **0.864** | **0.881** | **0.872** | **0.944** | **0.950** | **0.946** |

## 3.1.3 Open boundary

A proper location, far from the stations where the imposed water level is precisely defined, should be selected as the open boundary of the enclosed PG model. Applying two line boundaries, i.e., (1) connecting Sirik-Musandam at the Strait of Hormuz and (2) Jask-Almasnaeh in the west of Gulf of Oman, 1-hour model outputs are compared with measurements to choose the better location (Fig. 7). Considering the accuracy of predictions at all stations (Table 4), Jask-Almasnaeh is selected as the open boundary.

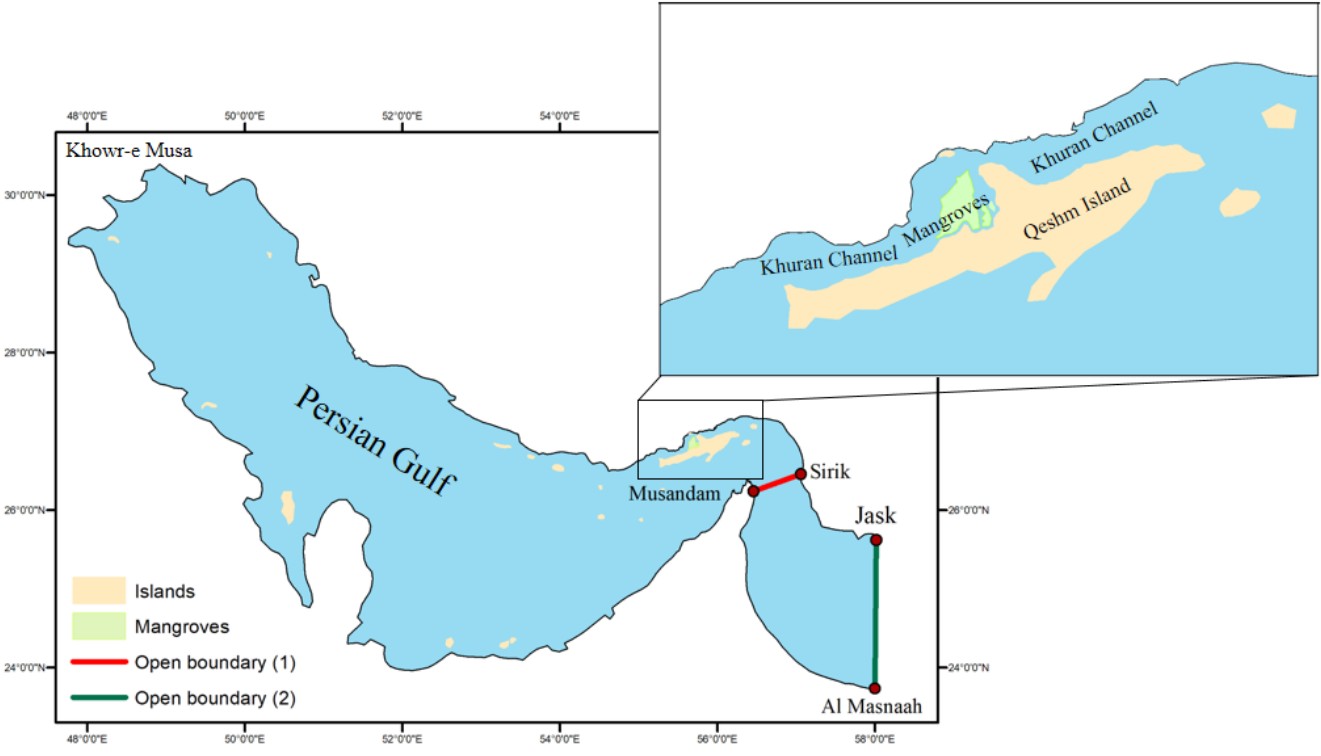

**Figure 7: Two alternative locations of open boundary of the model.**





**Table 4: Comparison of correlation coefficient for model results and observations in two different location of open boundary.**

| Station | RMSE | | $R^2$ | | Correlation | |
|---|---|---|---|---|---|---|
| Open boundary | 1 | 2 | 1 | 2 | 1 | 2 |
| Bustaneh | 0.11 | 0.1 | 0.95 | 0.96 | 0.98 | 0.98 |
| Chiruyeh | 0.13 | 0.15 | 0.89 | 0.85 | 0.96 | 0.94 |
| Shahid Rajaee | 0.15 | 0.16 | 0.97 | 0.96 | 0.98 | 0.98 |
| Khamir | 0.26 | 0.25 | 0.94 | 0.94 | 0.98 | 0.98 |
| Pohl | 0.59 | 0.29 | 0.65 | 0.91 | 0.91 | 0.97 |
| Larak | 0.32 | 0.15 | 0.81 | 0.95 | 0.94 | 0.98 |
| Taheri | 0.13 | 0.15 | 0.91 | 0.88 | 0.96 | 0.94 |
| Nayband | 0.14 | 0.15 | 0.90 | 0.87 | 0.96 | 0.94 |
| Dayyer | 0.15 | 0.17 | 0.91 | 0.88 | 0.96 | 0.94 |
| Dargahan | 0.27 | 0.34 | 0.91 | 0.87 | 0.96 | 0.93 |
| Bushehr | 0.25 | 0.25 | 0.67 | 0.67 | 0.94 | 0.93 |
| Genaveh | 0.11 | 0.11 | 0.95 | 0.95 | 0.98 | 0.98 |
| Deylam | 0.41 | 0.14 | 0.56 | 0.95 | 0.77 | 0.98 |
| Kharg | 0.28 | 0.07 | 0.63 | 0.97 | 0.81 | 0.99 |
| Lavar | 0.1 | 0.09 | 0.93 | 0.95 | 0.97 | 0.98 |
| Lavan | 0.08 | 0.07 | 0.95 | 0.96 | 0.98 | 0.98 |
| Farur | 0.1 | 0.1 | 0.94 | 0.94 | 0.97 | 0.97 |
| Sirik | 0.26 | 0.26 | 0.84 | 0.86 | 0.93 | 0.93 |
| Bahman | 0.30 | 0.26 | 0.84 | 0.88 | 0.92 | 0.94 |
| Basaidu | 0.29 | 0.26 | 0.83 | 0.87 | 0.92 | 0.93 |
| Kish Island | 0.16 | 0.17 | 0.87 | 0.84 | 0.94 | 0.91 |
| Kaveh | 0.26 | 0.37 | 0.94 | 0.88 | 0.97 | 0.94 |
| Al Ruwais | 0.14 | 0.26 | 0.90 | 0.68 | 0.95 | 0.85 |
| Mina Salman | 0.22 | 0.22 | 0.81 | 0.81 | 0.93 | 0.94 |
| Al Khiran | 0.3 | 0.23 | 0.7 | 0.83 | 0.84 | 0.92 |
| Kuwait | 0.44 | 0.48 | 0.84 | 0.81 | 0.96 | 0.96 |
| **Average** | **0.229** | **0.202** | **0.848** | **0.881** | **0.937** | **0.950** |

Following Ranji et al. (2016), TPXO global tide model is selected to define the water level at the open boundary (Egbert and Erofeeva, 2002). Thirteen tidal constituents ($M_2$, $S_2$, $N_2$, $K_2$, $K_1$, $P_1$, $Q_1$, $O_1$, $M_4$, $MS_4$, $MN_4$, $P_1$, MM, and MF) are used to simulate the tide elevations by TPXO8.

**3.2 Comparisons and validation**

Small water bodies such as marginal seas cannot produce a considerable response to astronomical tidal forcing. Sensitivity
analysis shows that the effect of tide generating forces in the 2D hydrodynamic modeling of the PG is limited to less than 1%. The sensitivity analysis confirms the small effect of the horizontal eddy viscosity. Here the Smagorinsky's formulation, with a default value of 0.28, is adopted in the numerical model. Flooding and drying are also included in the modeling for better simulation of shallow areas, particularly the Khuran Channel.

Figure 8 shows the comparisons between the time series of the simulated and measured water levels at 22 stations around the
PG (Fig. 2). A good performance of the model in reproducing water levels is observed, particularly in northern stations where the quality of employed bathymetry data is better. The maximum discrepancy is noticed at Kuwait station, where a considerable





difference between the amplitudes of simulated and measured water levels exists. Using 1-hour water levels at four sample stations, i.e., Bushehr, Nayband, Mina Salman, and Sirik, Fig. 9 shows the favorable statistical correlations of model outputs and data. Table 3 lists the statistical parameters of other stations.

The performance of the numerical model can also be evaluated using the major tidal constituents of Admiralty tide tables. Figure 10 shows the comparisons of the amplitudes of four main constituents of Admiralty observations (see Fig. 3) and model outputs, in which the corresponding values of the model have been extracted by the Admiralty method (Glen, 2015). The agreement is better for $M_2$ and $S_2$ in comparison to $K_1$ and $O_1$. Figure 11 shows comparison of measured and simulated velocity magnitudes (speed) at eight observation stations. The model is doing a reasonably good job considering the normal lower

accuracy of the simulation of current speeds, in comparison to water levels, in 2D hydrodynamic modeling.

**Figure 8: Comparison of simulated (solid-line) and measured (dash-line) water levels.**







**Figure 8: Comparison of simulated (solid-line) and measured (dash-line) water levels (continued).**




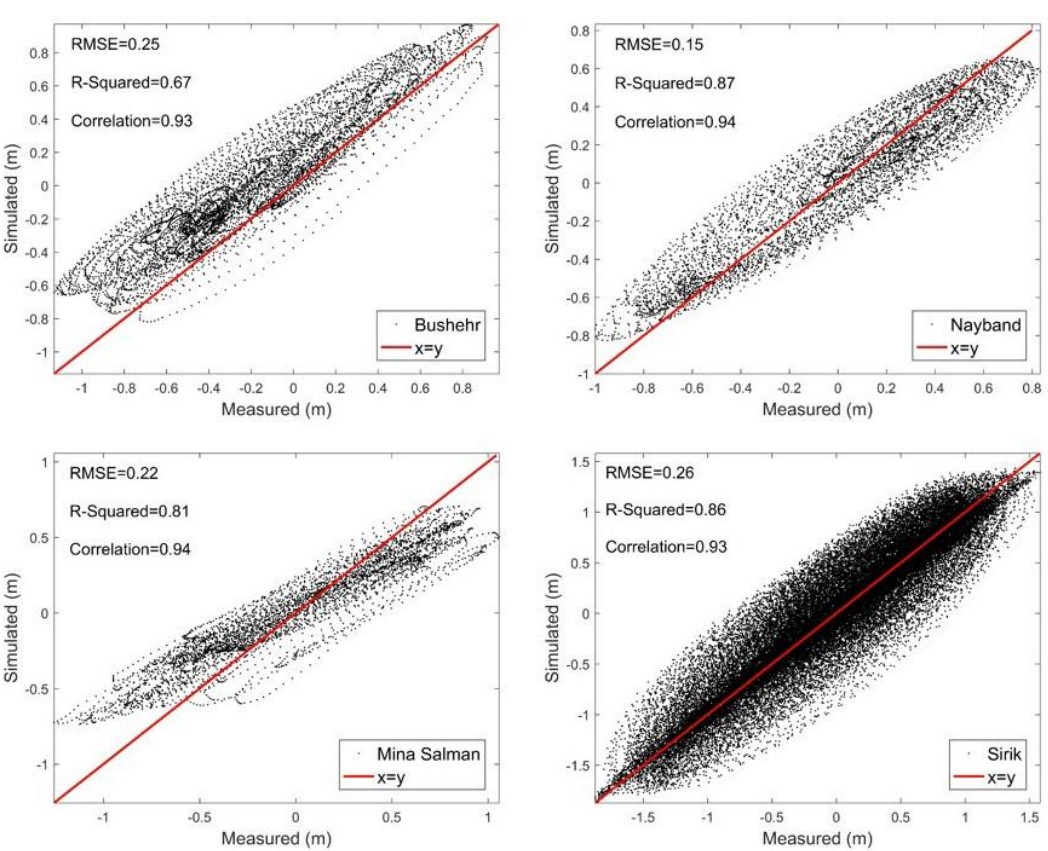

**Figure 9: Correlation analysis of simulated and measured water levels.**

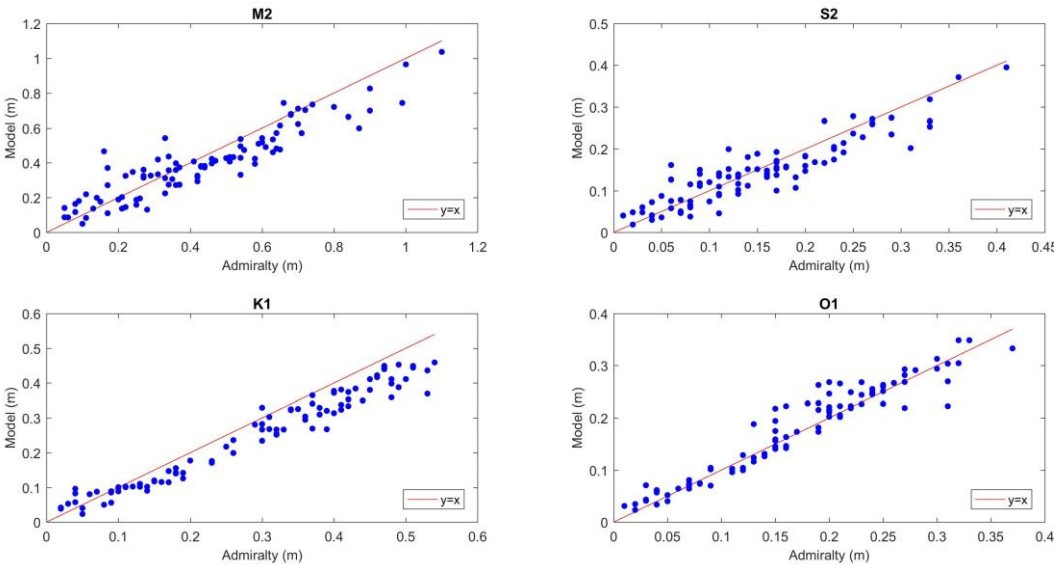

**Figure 10: Comparison of amplitudes of four principal constituents from Admiralty's observations and model results.**






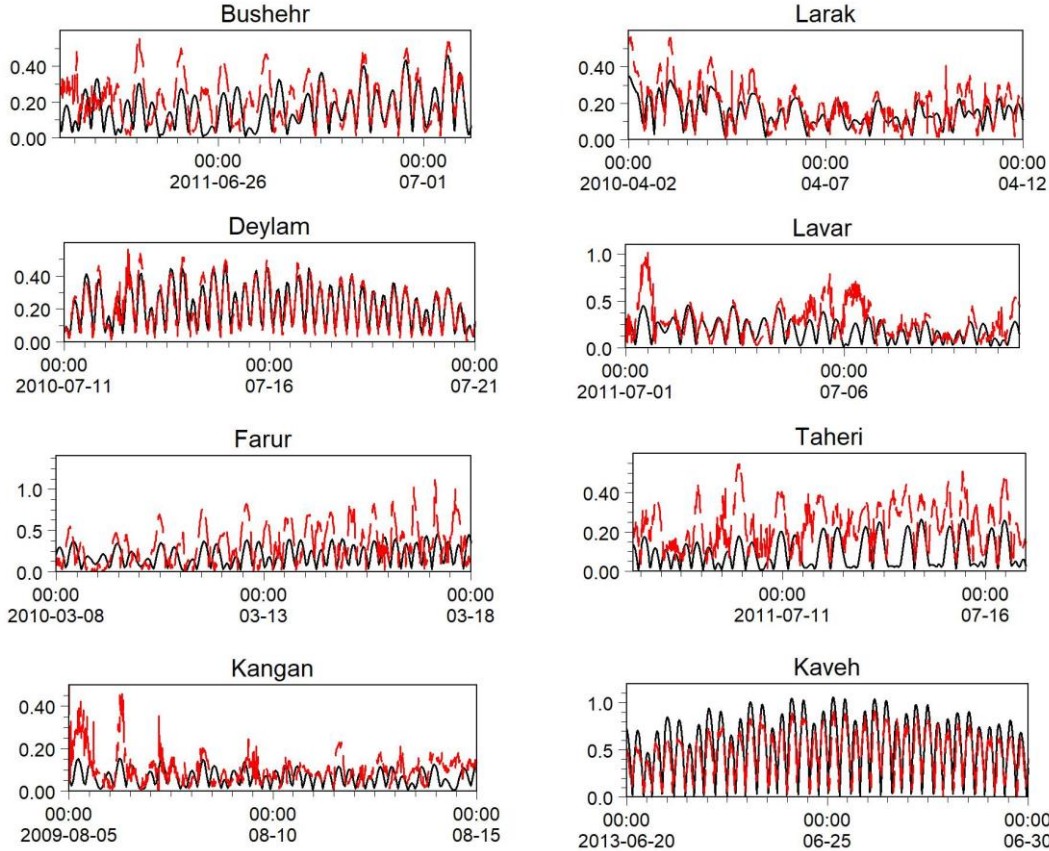

**Figure 11: Comparison of the simulated (solid-line) and measured (dash-line) current speeds.**

### 3.3 Model results

#### 3.3.1 Co-tidal and co-range charts

Tidal wave motions can be visualized in the form of co-tidal charts. Figures 12 to 15 present the co-tidal and co-range charts of four principal tidal constituents of $M_2$, $S_2$, $K_1$, and $O_1$, respectively. The tidal constituents are extracted from the modelled water levels by the IOS method (Foreman et al., 2009). Kriging interpolation technique, which makes predictions at unsampled locations using a linear combination of known value at nearby sampled locations, is used for the production of the charts (Oliver and Webster, 1990).

The tidal constituents (tidal wave) firstly propagate from the Gulf of Oman through the Strait of Hormuz and southwestward into the PG as a progressive wave (Fig. 1). They continue to propagate northwestward, i.e., the direction of the axis of the PG, as Kelvin waves around the coastlines and as progressive waves in the center of the PG until they reach the head of the PG, where reflection takes place. Semidiurnal and diurnal tidal waves in the PG can be represented by two Kelvin waves travelling

in opposite direction. It is observed that the waves rotate around a nodal point (AP) instead of oscillating about a nodal line

(discussed more in Sec.4.1) As the PG is located in the northern hemisphere, the tidal waves present a counter-clockwise rotation, which can be observed in co-tidal curves of $M_2$, $S_2$, $K_1$, and $O_1$ (dash-lines in Fig. 12 to 15).

The semidiurnal constituents of $M_2$ and $S_2$ have two APs, located in the northwest and southern ends of the PG, but the diurnal constituents, i.e., $K_1$ and $O_1$, have a single AP at the center part of the PG near Bahrain. The simulated co-tidal and co-range charts of the model generally agree with the corresponding Admiralty charts (UK Hydrographic Office, 1999), which are based

on direct observations over the sea and shallow waters (Doodson and Warburg, 1941). The APs location of Admiralty charts have also been plotted, to be compared with the derived APs from the numerical model.

Figure 12 and 13 show that the patterns of co-tidal and co-range charts of two semidiurnal constituents are similar. However, the amplitudes of $M_2$ are higher than the amplitudes of $S_2$ by a factor of two. High amplitudes of semidiurnal constituents are observed near the Strait of Hormuz and northwest of the PG, with the highest values at Khuran Channel.

Similarly, the diurnal charts of Fig. 14 and 15 are comparable to each other although the amplitudes of $K_1$ are larger than the amplitudes of $O_1$. The large amplitudes of these two diurnal constituents are observed in the northwest and south of the PG.

The change of amplitudes of constituents in Fig. 12 to 15 are comparable with the corresponding variations of diurnal and semidiurnal constituents along the north coastline of the PG, which can be observed from the contribution of tidal constituents in Fig. 4.


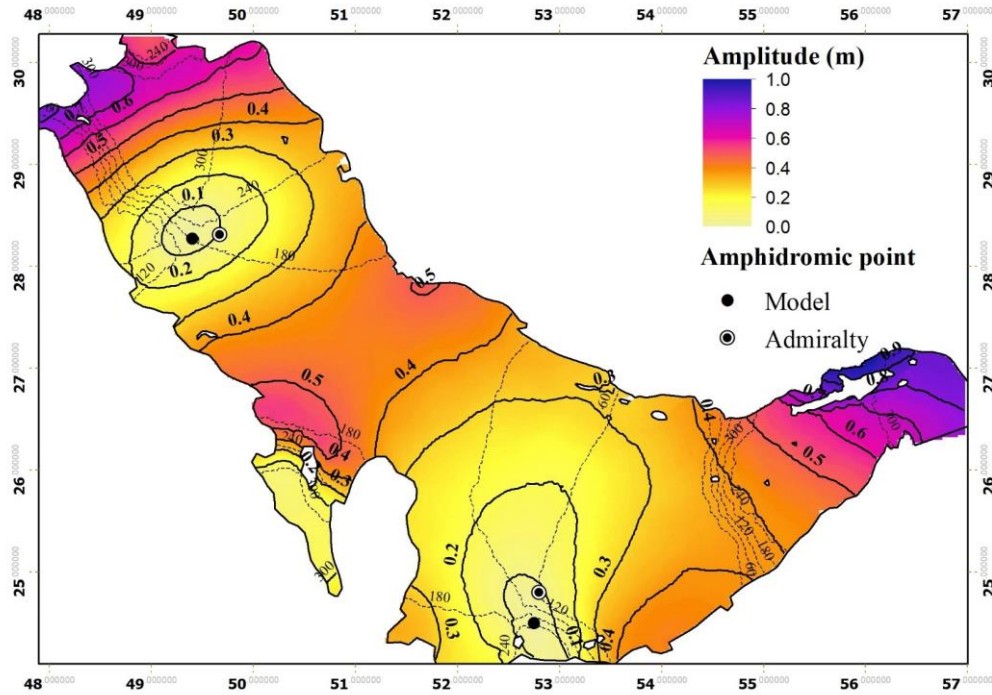

Figure 12: Simulated co-tidal and co-range charts of M₂. Solid lines denote co-amplitude lines and dash lines denote co-tidal lines.


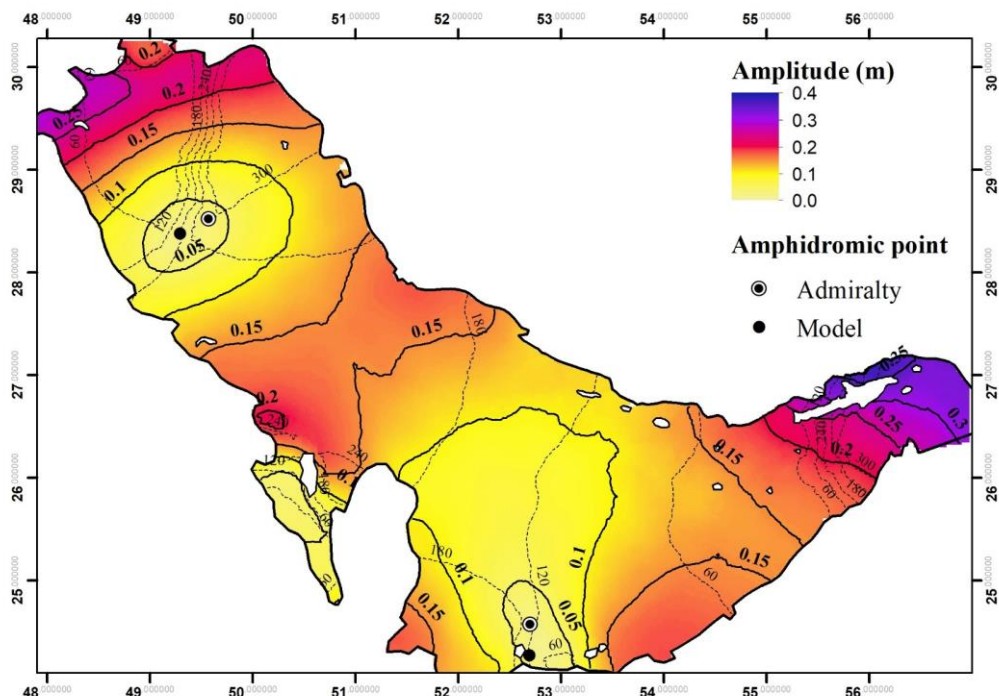

**Figure 13: Co-tidal and co-range charts of $S_2$. Solid lines denote co-amplitude lines and dash lines denote co-tidal lines.**

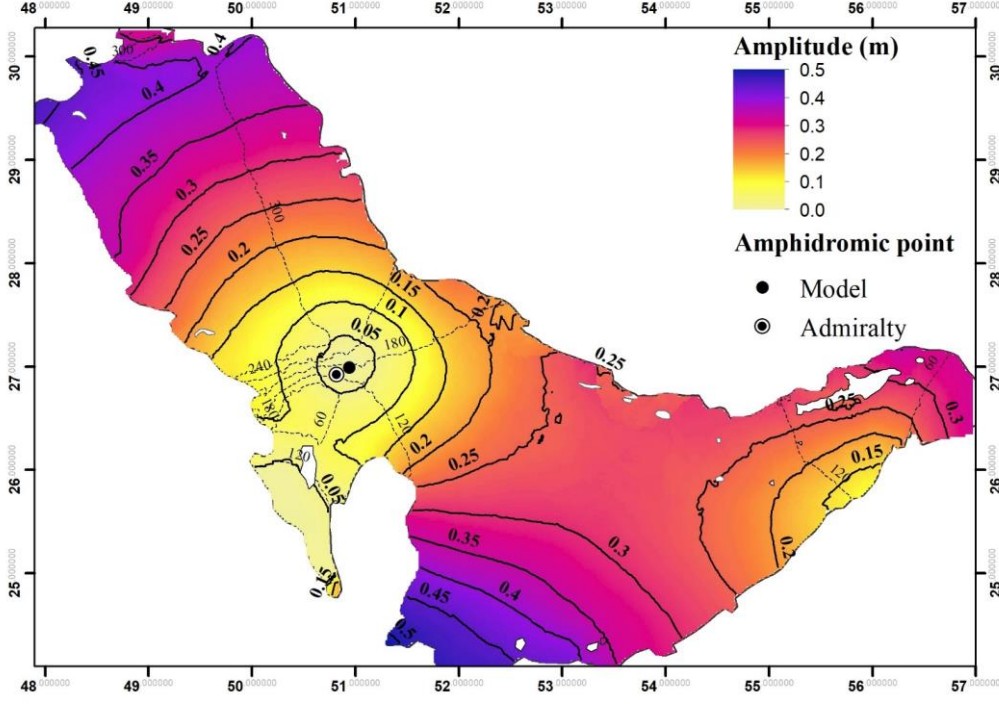


**Figure 14: Co-tidal and co-range charts of $K_1$. Solid lines denote co-amplitude lines and dash lines denote co-tidal lines.**




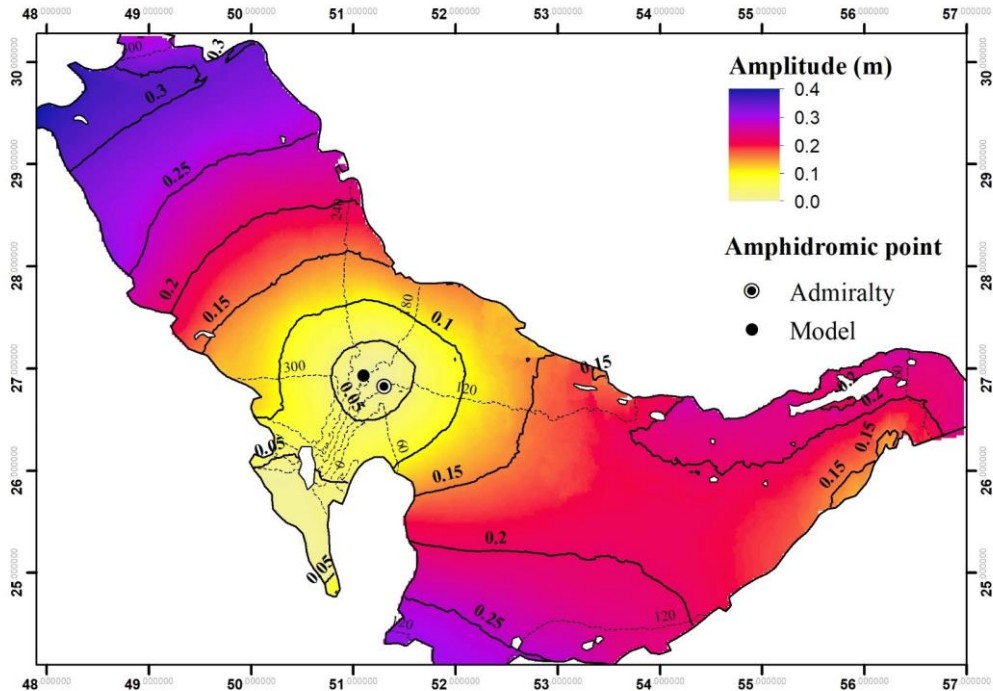

**Figure 15: Co-tidal and co-range charts of O$_1$. Solid lines denote co-amplitude lines and dash lines denote co-tidal lines.**

### 3.3.2 Shallow-water constituents

Shallow-water constituents are caused by the nonlinear distortions of the tides, in shallow water. Figure 16 presents the distribution map of the summation of modelled shallow-water constituents in the PG (Table 2). The locations of the higher shallow-water constituents are comparable with the bathymetry of the PG (Fig. 6), i.e. the shallower the water depth, the higher the shallow-water constituents. It is also observed that the map fully conforms to the contribution of shallow-water constituents in Fig. 4 As an example, because of the decrease of water depth the contribution of shallow-water constituents increase from

3.3% to 8.9% from Larak to Pohl stations. The maximum amplitude of shallow-water constituents in the PG is about 0.55 m at the Khuran Strait (Fig. 16).





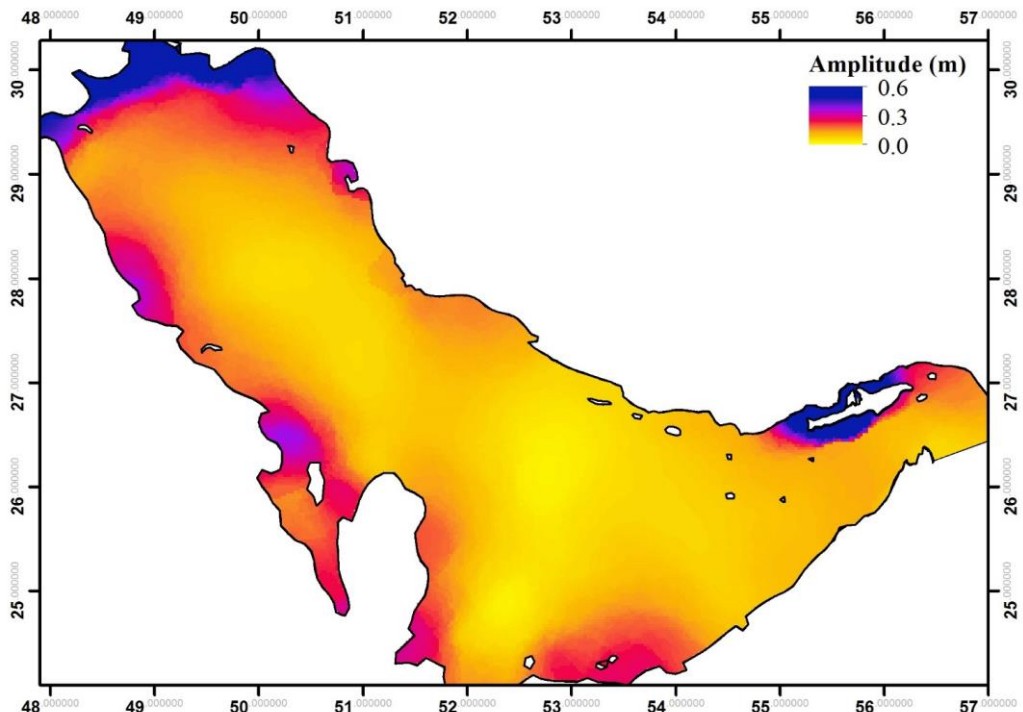

**Figure 16: Map of shallow-water constituents in the PG.**

### 3.3.3 Shape Factor

Figure 17 shows the distribution map of shape factor ($F = \frac{A_{K_1}+A_{O_1}}{A_{M_2}+A_{S_2}}$, $A$: constituent amplitude) over the PG, where the type of tide is generally mixed, mainly semidiurnal (0.25<$F$<1.5). This is in agreement with the results of the field measurements at the north coastline (Figure 4). It is also observed that the type of tide alters near the semidiurnal and diurnal APs.





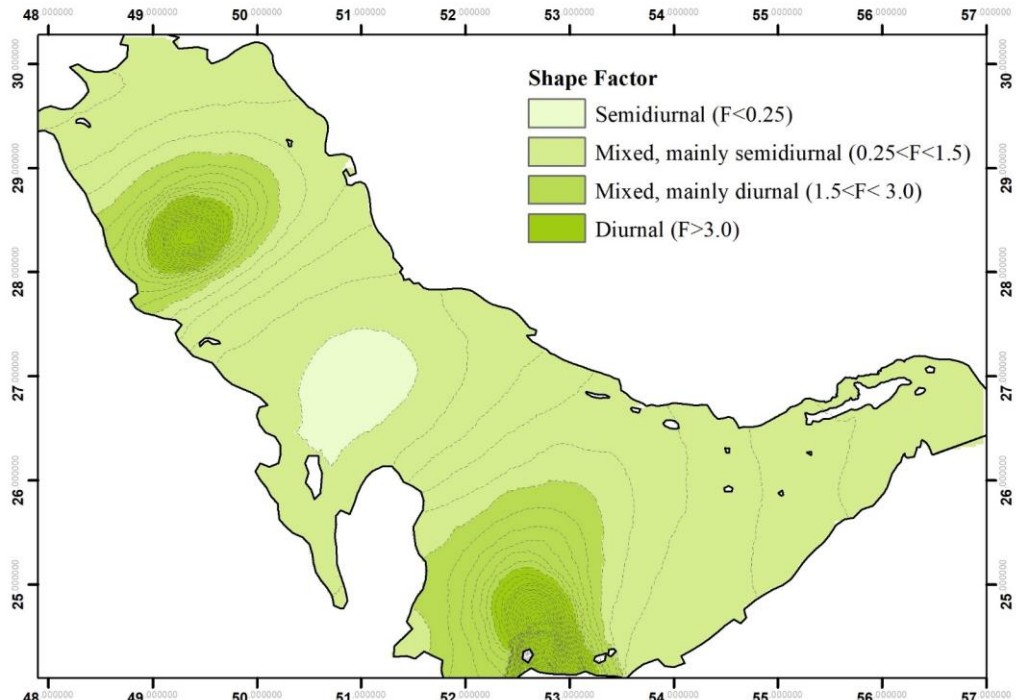

**Figure 17: Classification of tides in the PG.**

### 3.3.4 Resonance period


The effective period of resonance in the PG can be determined by its amplified response to various forcing periods. Assuming similar input conditions in the developed numerical model, the sinusoidal waves with the constant amplitude of 1 m are applied at the open boundary. An amplification factor is defined as the ratio between the wave height at the indicated location to the wave height at the open boundary. Seven scenarios of input waves with different wave periods are considered to represent the

response of tidal constituents to resonance (Table 5).

Figure 18 shows the amplification factor for input wave periods of 24, 12, and 6 hours in the PG. It is observed that the amplification factor is substantially higher in the Strait of Hormuz and Khuran Channel, northwest of the PG, and southeast of Qatar.

Defining three numerical stations at these areas (1, 2, and 3 in Fig. 18), Table 5 presents the amplification factor of all modeled

scenarios. The diurnal and  semidiurnal constituents show high amplification factors in three selected numerical stations and in $1^{st}$ and $2^{nd}$ stations, respectively, which is in accordance with the study of Mashayekh Poul (2016). The shallow-water constituents also resonate in some parts of the basin, particularly in Khuran Channel and Strait of Hormuz. It should be added although various factors affect the increase of the tidal elevation in the numerical model, comparing the changes of similar waves with different wave periods at checkpoints reveals that it is reasonable to relate the resonance phenomenon to the

amplification factor.





**Table 5: Tidal amplification factor for different input wave periods at numerical stations 1, 2 and 3.**

| Period (h) | Related constituents | Amplification factor | | |
|---|---|---|---|---|
| | | Station 1 | Station 2 | Station 3 |
| 26 | $Q_1, \rho_1$ | 1.25 | 1.54 | 1.21 |
| 24 | $K_1, O_1, P_1$ | 1.29 | 1.31 | 1.29 |
| 22 | $OO_1, SO_1$ | 1.32 | 1.27 | 1.14 |
| 12 | $M_2, S_2, L_2, N_2$ | 1.72 | 1.2 | 1.06 |
| 8 | $MK_3, SO_3, SK_3, 2MK_3 (MO_3)$ | 2.13 | 0.9 | 1.36 |
| 6 | $S_4, MK_4, MS_4, SN_4, M_4, MN_4$ | 2.71 | 0.62 | 0.84 |
| 4 | $2MN_6, M_6, 2MS_6, 2MK_6, 2SM_6$ | 2.84 | 0.37 | 0.49 |

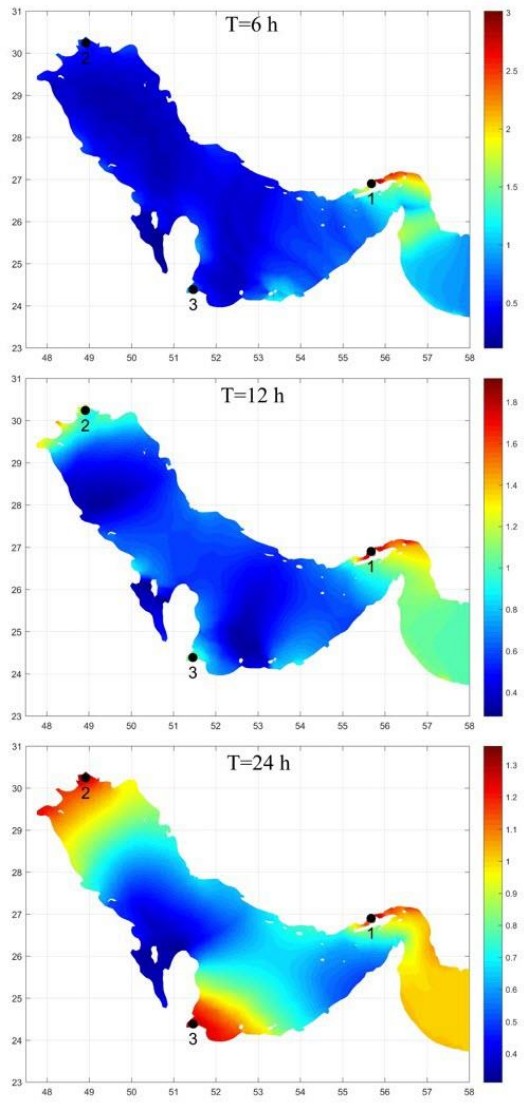

**Figure 18: The tidal amplification factors for the wave periods of 6 h (top), 12 h (middle) and 24 h (bottom).**





### 3.3.5 Maximum tidal range

As depicted in Fig. 19, the tidal wave has a different dynamic in shallow waters of the PG, in comparison to deep waters of Gulf of Oman. The wave amplitude increases up to about 2 m in Khuran Channel because of resonance and funneling due to the decrease of the width. These two phenomena are also responsible for the mild tidal amplification at southeast of Qatar in the southern coast of the PG. Moreover, the funneling and shoaling, due to a decrease in water depth, result in the high increase of tidal amplitude, between 1.65 m and 1.95 m, at Kuwait Bay and Khowr-e Musa in northwest of the PG.

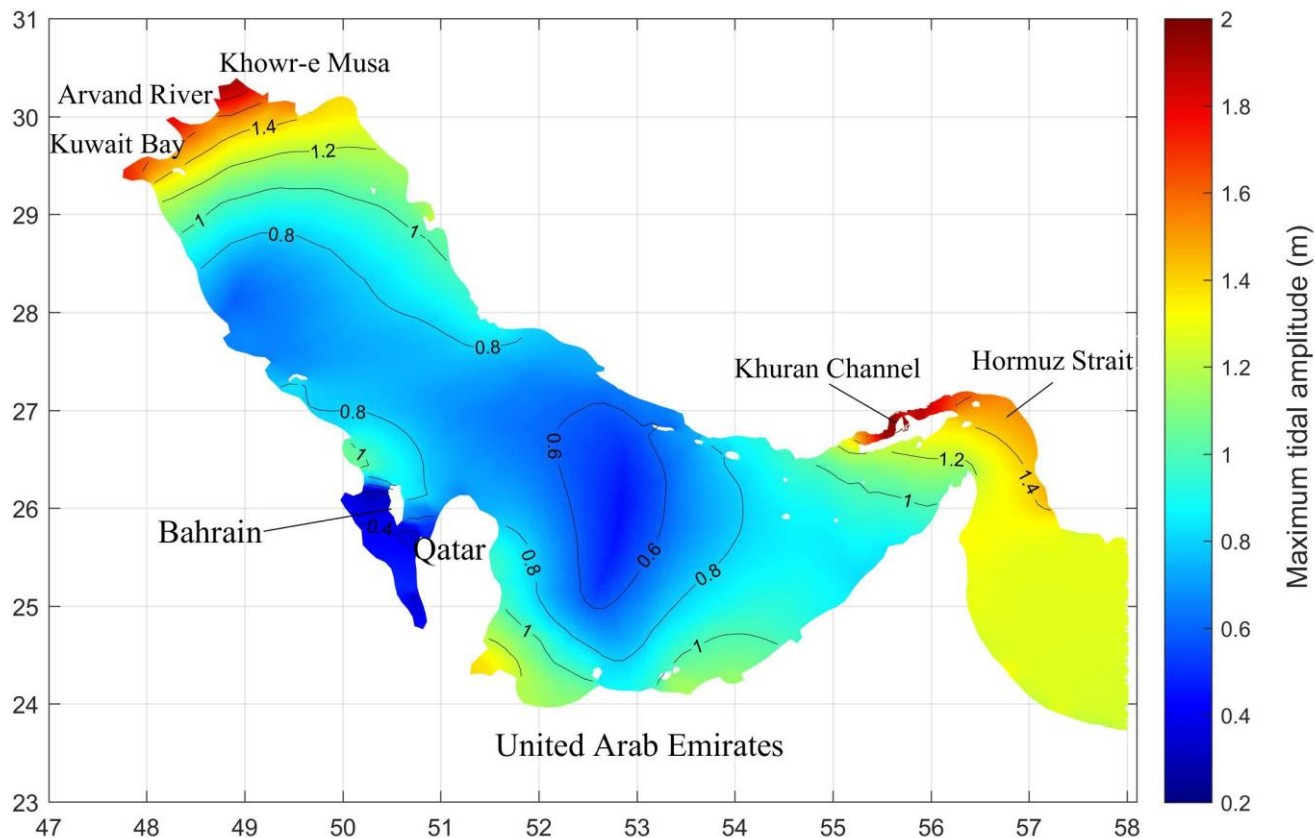

**Figure 19: Maximum tidal amplitude.**

### 4 Discussions on tidal wave propagation

The numerical model is used to study the details of the propagation of tidal wave in the PG. Three numerical experiments are designed to investigate the role of important influencing factors on tidal behavior, including the Coriolis force, friction, and bathymetry. The original model includes the real bathymetry of the PG, the Coriolis force and Manning roughness coefficient of 0.016 s m$^{-1/3}$ (i.e., $n=60$ m$^{1/3}$ s$^{-1}$). The Coriolis force is removed from the original model in the first numerical experiment.





In the second case, a horizontal flat bed with constant water depth of 36 m is assumed as the bathymetry of the PG. The
constant water depth is also used in the third numerical test with the Manning coefficient of 0.002 s m$^{-1/3}$, i.e., the smallest
value that does not make the model unstable, to concentrate on the effect of bottom friction.

### 4.1 Coriolis force

An amphidromic system develops as the result of the combined constraint of ocean basin geometry and the influence of the
Coriolis (Wright et al., 1999). The development of the amphidromic system in a gulf occurs when both standing and Kelvin
waves form.

If the gulf is sufficiently wide, in comparison to Rossby deformation radius (say $W > 2RR$, $W$; width of the gulf, RR; Rossby
deformation radius), the propagation of tide around the boundaries of a wide gulf is considered as a Kelvin wave since the
incoming and outgoing waves do not significantly overlap and interfere. On the other hand, rotation is insignificant in a narrow
gulf (say $W < RR$) and the standing wave is set up by reflecting the wave that entered from the open ocean (Hautala et al.,
2005; Vallis, 2006). The standing waves have nodes and antinodes, with zero and maximum amplitudes, respectively,
separated by a distance of $\lambda/4$, where $\lambda$ is the wavelength of the original progressive wave. The nodal line crosses the gulf at a
distance of $\lambda/4$ from its head (or $i\lambda/4$, i=3, 5, ...) (Pugh, 1987).

For an intermediate width, with respect to Rossby deformation radius, the tide will have the characteristics of both wide and
narrow gulfs. In this intermediate inlet, the tide will behave like the Kelvin wave on the boundaries. The amplitude of the
Kelvin wave increases in parallel to the wave crest because of the Coriolis acceleration, where in the northern hemisphere, the
amplitude increases to the right of the wave crest. The incoming wave from the ocean is thus amplified on the right towards
the coastline and the outgoing wave is similarly increased at the opposite coastline. The tide at the center axis of the gulf is
strongly similar to the case of a standing wave, i.e., low amplitude at the node and high amplitude at either side with an out of
phase relationship. The nodal line which extends across the channel when rotation is not important, turns into the nodal point
or AP in the center of the gulf around which the tide progresses (Hautala et al., 2005; Vallis, 2006).

Figure 20 shows the rotational effect of the Coriolis force on two hypothetical rectangular gulfs with intermediate width,
located in the northern hemisphere. It is observed that the Coriolis force results in one AP at $\lambda/4$ from the head of the gulf
($L < 3\lambda/4$, where $L$ is the length of the gulf), and two APs at $\lambda/4$ and $3\lambda/4$ from the head of the gulf ($L > 3\lambda/4$).




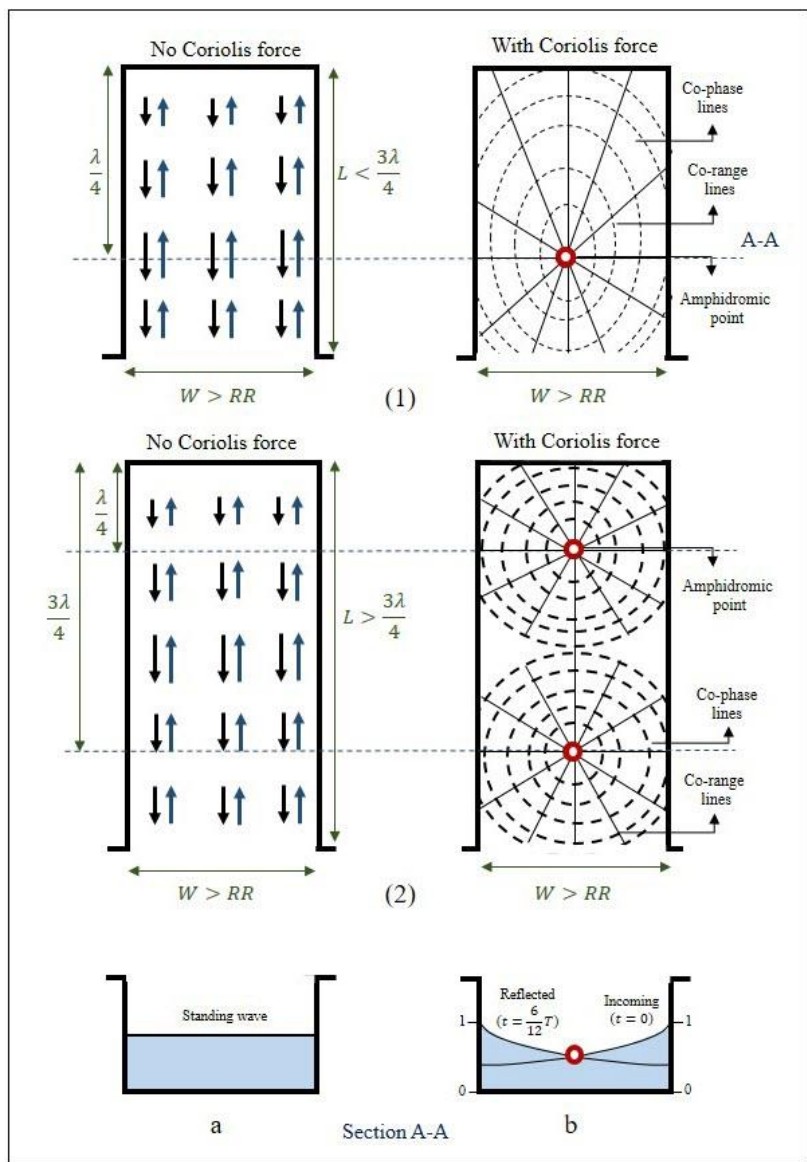

**Figure 20: Plan (cases (1) and (2)) and cross-section (bottom) views of a hypothetical rectangular gulf located in the northern hemisphere. The Coriolis force is excluded on the left side (similar to first numerical experiment). The right side is similar to original model, including the Coriolis force. Dash lines and solid lines in plan views denote co-range lines and co-tidal lines, respectively. In plan views, blue and black arrows show incoming and reflected tidal wave directions, respectively. The solid line and dash line in cross-section views respectively indicate the geometric locations of the crests of incoming and reflected Kelvin waves. Blank red circles represent the APs.**

The inertia period $T_i$ is 26.17 h ($= \frac{2\pi}{f}$, $f$; the Coriolis parameter$= 2\omega sin\varphi$, $\omega$; the angular velocity of the earth's rotation, $\varphi$;

the latitude$=$ the average latitude is equal to 27° N in the PG). The Rossby deformation radius ($RR = \frac{\sqrt{gh}}{f}$, $h$; water depth) is

284 km using the estimated average depth of the PG of 36 m. Thus, the semidiurnal and diurnal tidal periods in the PG are in





order of the inertial period (26.17 h), and the maximum width of the PG (about 338 km) is more than the Rossby deformation
radius (but not much more, i.e., an intermediate case).

Figure 21 and 22 show the effect of Coriolis force on co-tidal charts of diurnal and semidiurnal constituents in the PG,
respectively. It is observed that the Coriolis force converts the nodal lines into nodal points, which results in forming
amphidromic systems for both semidiurnal and diurnal tides. Table 6 presents the wavelengths of the major tidal constituents

in the PG. Considering the PG length of about 1000 km, the modeling results in Fig. 21 and 22, i.e., one node (AP) for diurnal
constituents and two nodes (APs) for semidiurnal constituents, agree with the above theoretical discussion on rectangular gulfs
(Fig. 20).

**Table 6: Wavelengths of tidal constituents and locations of nodal lines.**

| Constituent | Period (h) | $\lambda = \sqrt{gh} \times T \ (km)$ | $\dfrac{\lambda}{4}$ | $\dfrac{3\lambda}{4}$ |
|---|---|---|---|---|
| $M_2$ | 12.42 | 840 | 210 | 630 |
| $S_2$ | 12 | 812 | 203 | 609 |
| $K_1$ | 23.93 | 1596 | 399 | 1197 |
| $O_1$ | 25.82 | 1722 | 430 | 1290 |


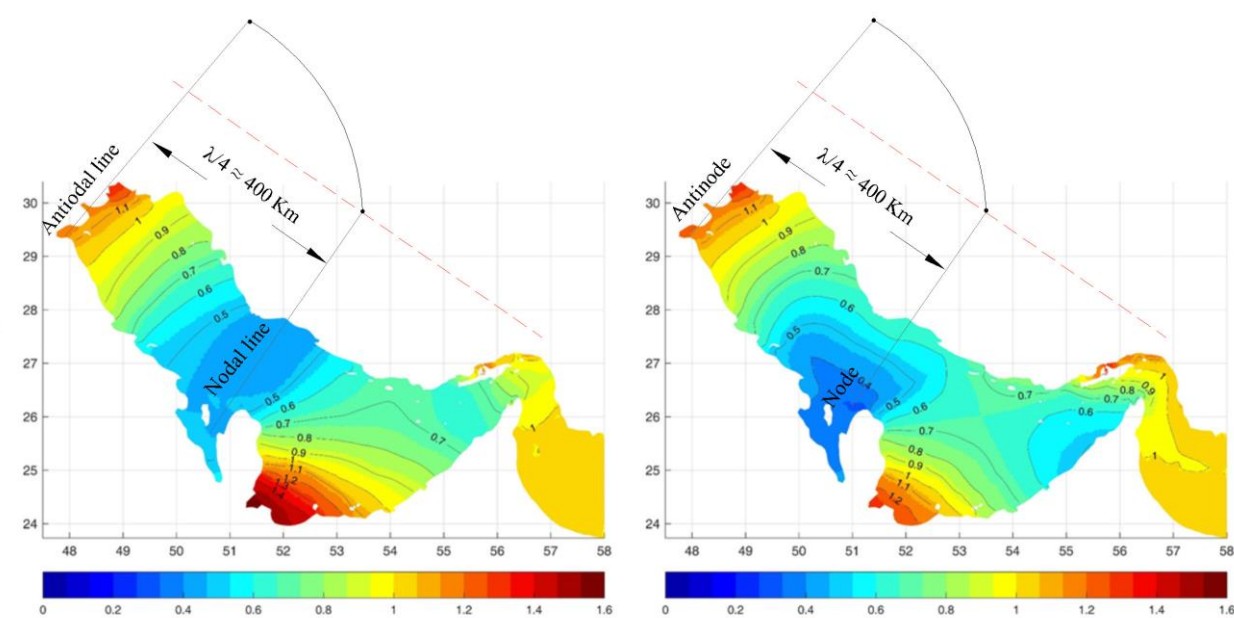

**Figure 21: The locations of the nodal line and node for diurnal constituents with the Coriolis force (right) and excluding the Coriolis force (left).**

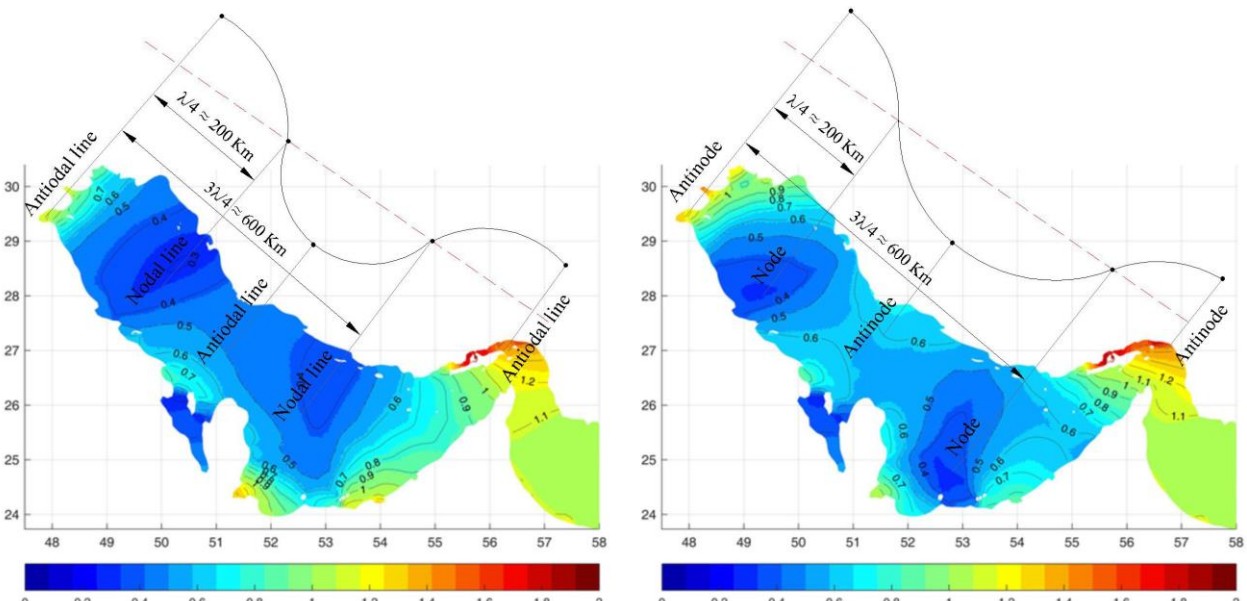

**Figure 22: The locations of the nodal lines and nodes for semidiurnal constituents with the Coriolis force (right) and excluding the Coriolis force (left).**

**4.2 Bathymetry**

The second numerical test aims to study the effect of bathymetry and water depth differences on tidal wave transformation in the PG. The bathymetry of the PG can be comprises a shallow zone at the closed end that extends along the southern part of

the basin, resulting in a highly asymmetrical cross-sectional longitudinal and transverse depth profiles (Fig. 6).

Figure 23 and 24 respectively present the numerical results of semi-diurnal and diurnal constituents on a hypothetical bathymetry with constant water depth of 36 m, in comparison to the original model with real bathymetry. The background color shows the change of amplitudes, i.e. the results of employing hypothetical bathymetry minus the original outputs. Co-tidal lines and the locations of the APs are also presented. The different location of co-tidal lines in this test case can be related

to the changes of tidal wave speed as it travels faster in larger water depths, e.g. vicinity of the Strait of Hormuz. An obvious shift of semidiurnal APs in the longitudinal direction of the PG is also observed (Fig. 23). The shifts of real southern semidiurnal APs of the PG towards the open boundary, in comparison to the test model, can be explained by the local changes of tidal wavelengths. The differences of locations are greater for the second southern APs because of the significant longer wavelengths in that region, in comparison to the test model. It is also observed that the average water depth of 36 m in this

numerical test, and the consequent shorter wavelength in deeper waters, has resulted in the appearance of third AP of $S_2$ in the southeast of the PG.

The transverse movements of the single diurnal APs in Fig. 24 can also be explained by the tidal wave propagation in the northern hemisphere. The shape of the PG between the head and AP of diurnal constituents can be approximated to a rectangular basin, where the upper half (north) is deeper than the lower half (south). Due to Coriolis acceleration in the northern





hemisphere, the incoming Kelvin wave is bounded to the right of the direction of motion and increases in amplitude parallel to the wave crest toward the right, i.e., the coastline of the north deeper part of the PG. Similarly, the reflected Kelvin wave follows the coastline of the south shallower part of the PG and increases its amplitude toward the southern coastline. Since the incoming and reflected Kelvin waves are banked up on the northern and southern sides of the PG, respectively, they are affected by the larger depth of the northern half and lesser depth of the southern half, respectively. The associated change of wavelength

thus leads to a decreased amplitude in the deeper part of the basin and an increased amplitude in the returning tidal wave in the shallower part, considering the wave energy conservation. On the other hand, the amplitude decreases by factor $\exp\left(-\frac{fy}{\sqrt{gh}}\right)$ (Pugh and Woodworth, 2014). Assuming the constant Coriolis parameter (*f*) because of the slight change in latitude in the cross section of the PG, the reflected wave should have a faster decrease of amplitude away from the coast than the incoming wave.

Figure 25 shows the aforementioned explanations in a hypothetical rectangular gulf located in the northern hemisphere. Comparing with the constant water depth, this pushes the AP away from the centerline into deeper north parts in the original model of the PG (Fig. 24). These results are in accordance with conclusions derived from an idealized process-based model for semi-enclosed basins by Roos & Schuttelaars (2011).

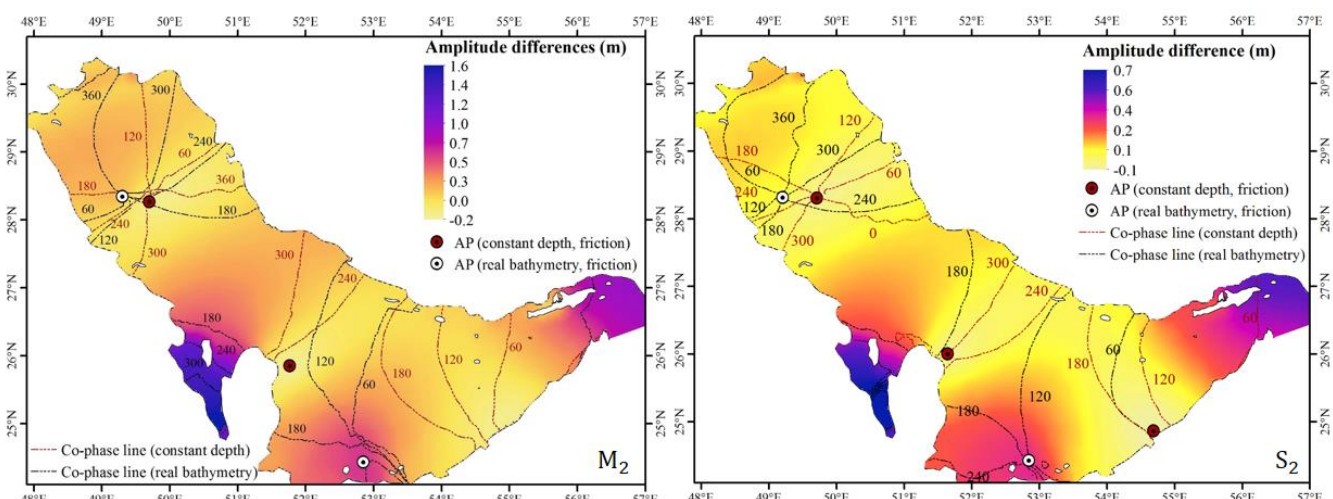

**Figure 23: Co-tidal chart of M₂ (left) and S₂ (right) constituents derived from model results in two scenarios including the constant water depth of 36 m and real bathymetry of PG. The background colors show amplitude differences in the two tests.**





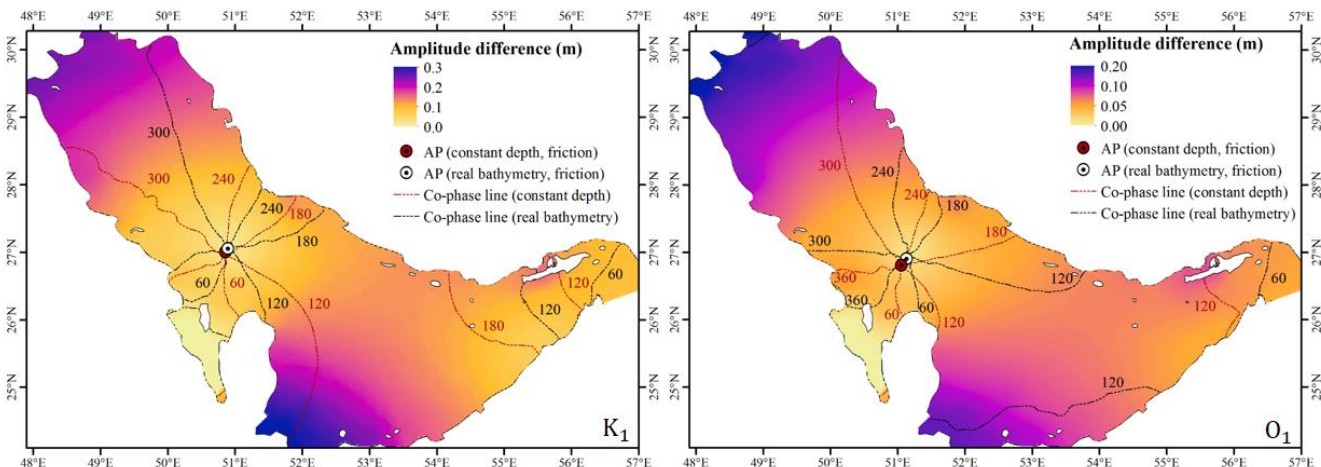

**Figure 24: Co-tidal charts of K₁ (left) and O₁ (right) constituents derived from model results in two scenarios including the constant water depth of 36 m and real bathymetry of PG., respectively. The background colors show amplitude differences in the two tests.**

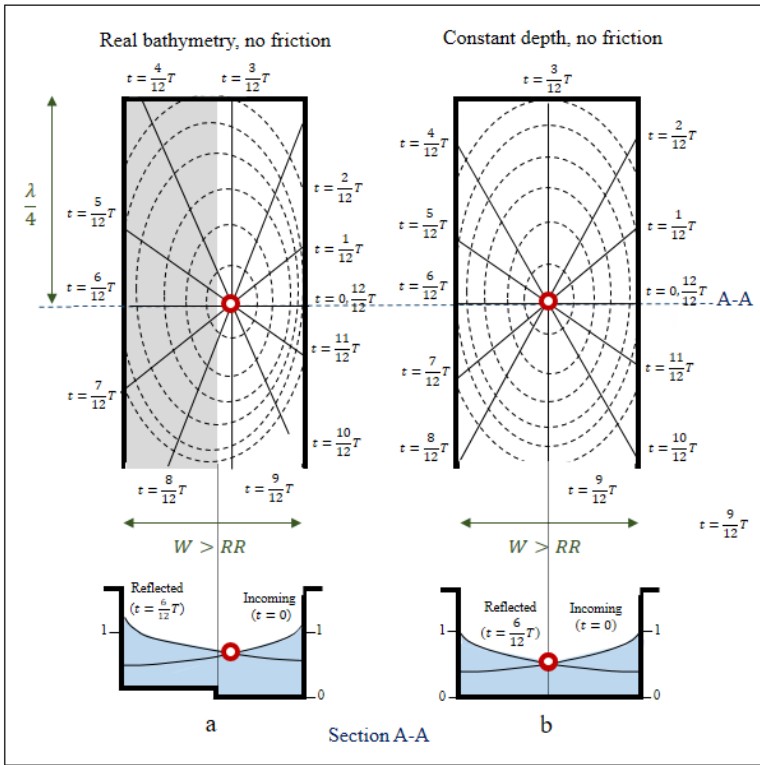


**Figure 25: Plan and a cross-section view of a hypothetical rectangular gulf located in the northern hemisphere. (a) Shallow zone (shaded in grey) is in the left side of the gulf. (b) Constant water depth. Dash lines and solid lines denote co-range and co-tidal lines in plan views, respectively. In cross-section views, solid lines and dash lines indicate the geometric location of Kelvin wave at t=0 and t=6/12T and blank red circles show the Aps, respectively.**





### 4.3 Friction


The effect of friction is examined in the third numerical test, modeling the tidal wave propagation on the hypothetical constant depth of the PG (second test) with and without bed roughness. The primary effect of friction is the exponential damping of Kelvin wave amplitude in the direction of tide propagation (Rienecker and Teubner, 1980; Roos and Schuttelaars, 2009). Furthermore, the reflected wave also dampens by the incomplete reflection from the head of the gulf, caused by the dissipation

in shallow water (Allen, 2009).

The incoming progressive (Kelvin) wave has a larger amplitude on the right-hand side of gulf in the northern hemisphere. The Coriolis force also results in the greater amplitude of reflected wave on the left-hand side of the gulf. As the reflected wave is weaker than the incoming wave due to dissipation damping, the APs are shifted from the centerline to the left side of the gulf. Figure 26 illustrates this phenomenon in a hypothetical rectangular gulf, located in the northern hemisphere.

Figure 27 and 28 present modeling results of the propagations of $M_2$, $S_2$, $K_1$, and $O_1$ constituents. It is observed that the APs move to the right side of the PG when the bottom friction is removed from the model. These results are in accordance with conclusions of Roos & Schuttelaars (2011), derived from an idealized process-based model for semi-enclosed basins. However, no displacement of the APs is observed in longitudinal axis of the PG.

Figure 27 and 28 show the increase of tidal amplitude all over the PG in modeling without friction. The change of tidal wave

height is larger in shallow areas, such as the northwest of the PG and south of Qatar and Bahrain, as well as narrow Strait of Hormuz.





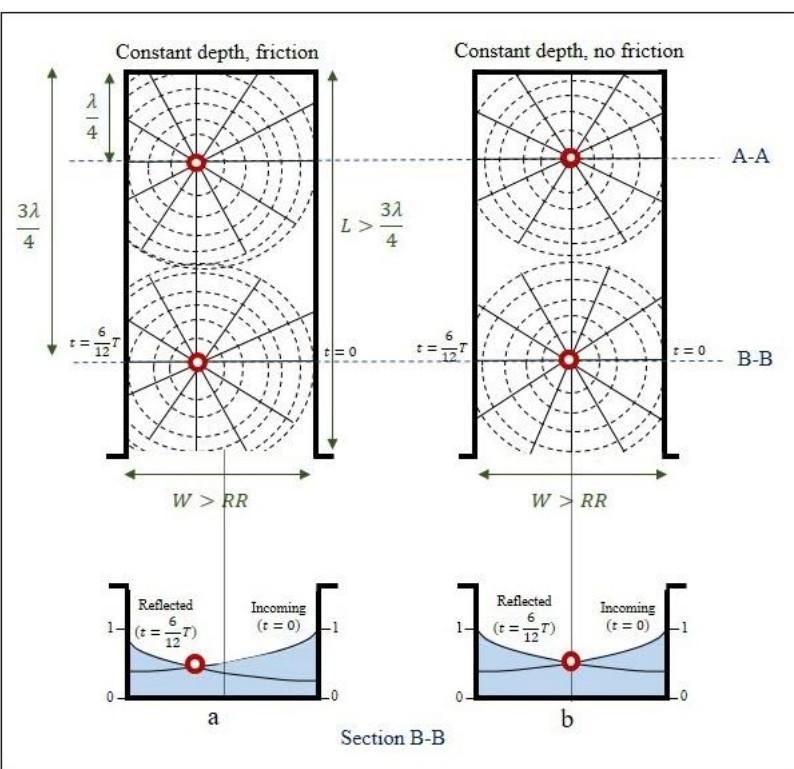

**Figure 26: Plan and a cross-section view of a hypothetical rectangular gulf, located in the northern hemisphere. (a) Constant depth with bottom friction. (b) Constant depth excluding the bottom friction. Dash lines and solid lines denote co-range lines and co-tidal lines, respectively. In cross-section views, solid lines and dash lines indicate the geometric location of Kelvin wave at $t=0$ and $t=6/12T$, respectively and blank red circles show the APs.**

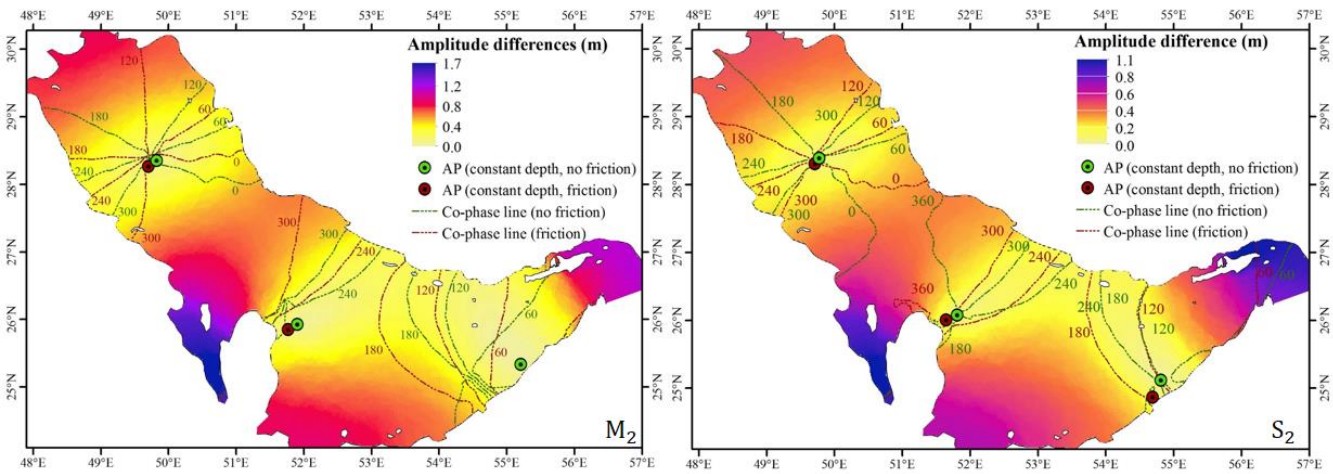

**Figure 27: Co-tidal charts of $M_2$ and $S_2$ constituents with constant water depth of 36 m in two scenarios with and without bed friction. The background colors show the differences of tidal wave amplitudes in the two cases.**



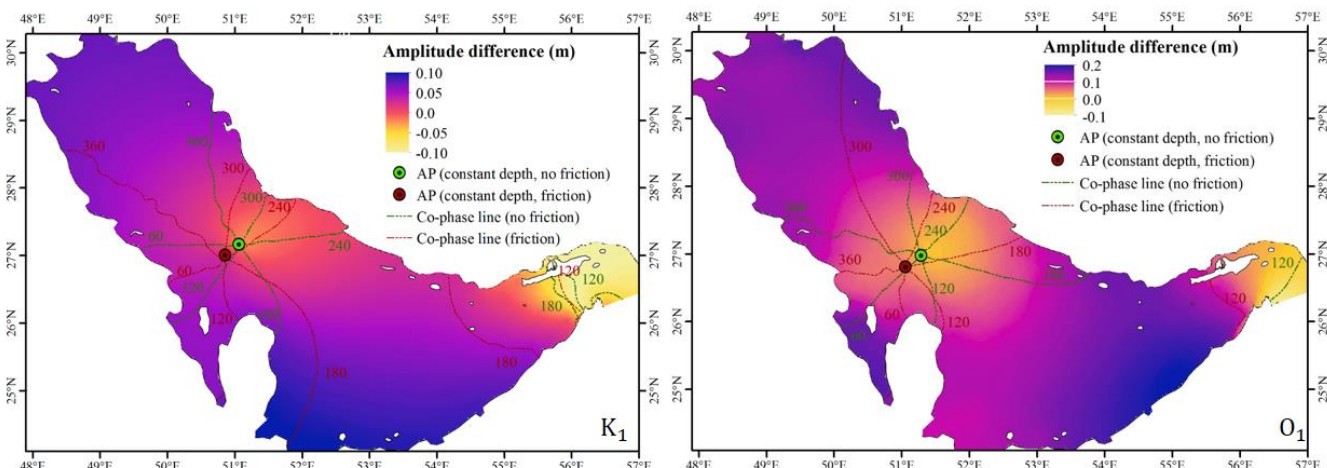

**Figure 28: Co-tidal charts of K₁ and O₁ constituents with constant water depth of 36 m s in two scenarios with and without bed friction. The background colors show the differences of tidal wave amplitudes in the two cases.**

## 5 Summary and conclusion

A 2D hydrodynamic model with a relatively high grid resolution, forced by thirteen tidal constituents at its open boundary, was set up to study the tidal wave propagation in the PG. Comparisons of the simulated water levels and tidal currents with measured data at the stations around the PG showed good model performance. The derived constituents from the model results were comparable with the observations of the UK Hydrographic Office (Admiralty) and shallow water constituents were similar to the extracted constituents out of measurements. The co-tidal and co-range charts of four principal tidal constituents

were also comparable with the corresponding Admiralty charts.

The results revealed that dominant tide in the PG is mixed mainly semidiurnal with the maximum tidal amplitude of about 2 m in the Khuran Channel and Khowre-Musa (Fig. 7). Considering the data of all northern stations, the largest contributions are semidiurnal, followed by diurnal, long-term, and shallow-water constituents, respectively. Series of numerical tests were carried out to study the effects of the governing factors, i.e., geometry and bathymetry, Coriolis force, and bed roughness on

tidal wave propagation in the PG. It was found that the Coriolis force, combined with constraint of the geometry, result in the rotation of Kelvin waves about the nodal point (AP) and developing the amphidromic system. The second test showed that a transverse step in bathymetry leads to different wavelengths of the incoming and reflected Kelvin waves, resulting in the increased amplitudes in shallow areas, which leads to the shift of diurnal APs toward the deeper part of the PG. The third numerical test demonstrated that bottom friction in a gulf results in the exponential decay of the incoming and reflected Kelvin

waves, moving the AP from the centerline.

The tide in the PG, which is the result of co-oscillation with the tide in the Gulf of Oman, can be represented by two Kelvin waves traveling in opposite directions. The tidal regime is affected by the Coriolis force, resulting in the generation of two



amphidromic systems for $M_2$ and $S_2$ and one amphidromic system for $K_1$ and $O_1$. Numerical simulations and observational data reveal that the natural oscillation period of PG is more or less diurnal.

More field measurements in the southern coast of the PG will be beneficial to improve the calibration and validation of the present model. Including the wind force and the discharge of major rivers, in particular Arvand River at the northwest corner of the PG, also increases the accuracy of the modeling, which is particularly important for the hindcast of water elevations and currents across the PG. Moreover, in spite of shallow water depth of the PG that validates 2D modeling, the future extension of present model to 3D will result in better simulation of hydrodynamics at Strait of Hormuz, with its pronounced vertical
salinity stratification, as well as the circulations near the land boundaries.

**Data availability**

The PMO and NCC provided us with the water level/current speed observational data and hydrography maps, respectively. Both are national data which are not publicly accessible. They would be accessed by direct request from the owner. The PMO website is available through https://www.pmo.ir/en/home and NCC website is available through https://www.ncc.gov.ir/en/.
The tidal constituent's extraction code is available through https://github.com/CADWRDeltaModeling/vtide (Foreman et al., 2009).
Two-minute gridded global relief for both ocean and land areas (ETOPO2v2) are available through https://www.ngdc.noaa.gov/mgg/global/relief/ETOPO2/ETOPO2v2-2006/ (NOAA, 2006).
Admiralty tide table volume 3 is available at https://wiac.info/docview.

**Author contributions**

This study was conducted as the master thesis of SMH under the supervision of MS. SMH did simulations, analysis, visualization and drafted the manuscript. MS participated in the interpretation of the results and prepared a critical revision of the manuscript.

**Competing interests**

The authors declare that they have no conflict of interest.

**Acknowledgment**

The authors acknowledge the PMO for providing tide measurements along the north coast of the PG. We are grateful to NCC for hydrography maps and observation data. Thanks are also extended to Ms. Zahra Ranji, Ph.D. candidate at K. N. Toosi University of Technology for her technical supports.





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
