# Peer review of "Tide characteristics and tidal wave propagation in the Persian Gulf"

_Ocean Science, 2021_

## Author Comment (AC1)

**Reviewer 1**

*The authors appreciate the reviewer's evaluation and detailed valuable comments. We tried our best to address all the raised issues and revise the paper accordingly. What follows is a point-by-point reply to the comments and the proposed modification. To facilitate following the comment sheet, comments and the responses are printed in blue font and black italic font, respectively. We hope the reviewer accepts the revisions and supports our revised paper merits the review process in Ocean Science.*

*Yours sincerely,*

*S. Mahya Hoseini*

*Mohsen Soltanpour*

1. This paper is a very basic study of the tidal characteristics of the Persian Gulf, which have been studied in great detail many times before. The authors do not convince me that their approach is more novel than the detailed study of others such as Ranji and Soltanpour (2021), who included spatially varying friction, as a function of water depth, mean velocity, vegetation, and bed sediment size. The authors claim that "Despite all past efforts, tidal modeling in the PG still needs to be improved, with comparisons to new water levels and current speeds measurements at different locations." I don't think that new observational data warrant another tidal study of the Persian Gulf, unless the authors can prove that their model performs better than previous models, which is missing. On the other hand, most data used are 10 years or older, so I don't understand what new data the authors are referring to. Due to the lack of novelty and scientific advance, I cannot recommend the publication of this paper.

*Thank you for your comments. We agree that the literature shows valuable studies on tidal modeling of the PG, e.g. Sabbagh-Yazdi et al. (2007), Pous et al. (2012), Ranji and Soltanpour (2021). However, the goal of the present research was to study the physics of tide in the PG. As we wrote in the first sentence of the abstract (i.e., 2D hydrodynamic model is employed to study the characteristics of tidal wave propagation in the Persian Gulf), the hydrodynamic model was used as a tool alongside other methods such as analysis of field measurements and propagation of Kelvin wave.*

*We acknowledge that our employed model has similarities with the numerical model of recently published study of Ranji and Soltanpour (2021). They proposed a general method, which was tested in the PG as a case study, for the spatial optimization of the Manning coefficient, as a function of water depth, mean velocity, vegetation, and bed sediment size, to increase the performance of 2D hydrodynamic modeling. Fig. $R_1$ presents the final results of their optimization process, which resulted in slight overall improvement of model outputs of about 3%, compared to applying a constant friction. It is observed that although a maximum improvement of 27.5% is obtained at Khuran Channel, the accuracy of predictions decreased at the southern PG coastline and near the islands, e.g., - 8% changes at Bahrain Station. The poor performance at the shallow water stations was related to the lack of good quality bathymetry data.*

[Figure]

*Fig. $R_1$: Changes of objective functions at different stations for spatially varying bottom friction versus the fixed Manning number (Ranji and Soltanpour, 2021).*

*It should also be added that unlike Ranji and Soltanpour's (2021) model that was only verified by water level measurements, here the numerical model was also compared with a large sets of tidal constituents of UKHO, and current speeds at different locations. In spite of above facts, as our*

*model does not prove to be superior to Ranji and Soltanpour's (2021) model, it is better to omit the above mentioned sentence, i.e. "Despite all past ...." in the revised manuscript.*

*The verified model was then employed to study the tidal wave characteristics and its propagation in the PG. This is the main objective of present study, which is not seen in past studies. The analysis of field data at north coastline of the PG is also new in this research (figure 4, page 7). A series of numerical tests was also conducted to investigate the various effects of PG geometry, PG bathymetry, Coriolis force, and bed friction on tidal wave propagation (Section 4: Discussions on tidal wave propagation, pages 23-32).*

*The following list can be considered as the innovations of the present research:*

- *Study the effects of the governing factors, i.e., geometry and bathymetry, Coriolis force, and bed roughness, on tidal wave propagation in the PG by series of numerical tests (section 4, pages 23-32).*
- *The tide in the PG was represented by two Kelvin waves traveling in opposite directions, as the result of co-oscillation with the tide in the Gulf of Oman (section 3.3.1, pages 16-17).*
- *Study of the effect of the Coriolis force on the tidal regime in the PG, which results in the generation of two amphidromic systems for $M_2$ and $S_2$ and one amphidromic system for $K_1$ and $O_1$ (section 4.1, pages 24-26).*
- *Study of natural oscillation period of PG (mainly diurnal) based on numerical simulations (section 3.3.4, pages 21-22).*
- *Presenting the map of tidal constituents on the northern coastline of the PG (figure 4, page 7), based on field measurements and the maps of shallow-water constituents (figure 16, page 20) and maximum tidal amplitudes (figure 19, page 23) in the PG, based on model results.*

  *The following sentence was omitted from the manuscript.*

  "Despite all past efforts, tidal modeling in the PG still needs to be improved, with comparisons to new water levels and current speeds measurements at different locations."

2.1. The model description is incomplete. Bottom friction treatment is not described.

*Thank you for the comment. The following paragraph will be added to the manuscript to define the treatment of bottom friction (page 10, line 145):*

Bottom stress in Flow Model (FM) module of MIKE 21 is determined as quadratic friction law ($\frac{\overrightarrow{\tau_b}}{\rho_w} = c_d \overrightarrow{u_b}|\overrightarrow{u_b}|$, $c_d$: drag coefficient, $\overrightarrow{u_b}$: depth-average velocity, $\rho_w$: water density) and the drag coefficient can be obtained from Manning number ($c_d = \frac{g}{(Mh^{\frac{1}{6}})^2}$, $M$: Manning number, $h$: water depth) (DHI, 2012). Proper determination of Manning number is important for the accurate hydrodynamic modeling.

2.2. Boundary conditions are also not properly described. What conditions are used for transport components? Do the tidal amplitudes vary along the open boundary?

*The open boundary and boundary condition were briefly discussed in page 12, lines 170-172.*

*The boundary condition in the MIKE 21 Flow Model can be chosen between two combinations of boundary input:*

1. *Specification of water level and the direction of the flow.*
2. *Specification of a flux boundary as either discharge, flow flux, or Rating curve through the boundary and the flow direction.*

*Following the first combination, the water level was imposed along the line open boundary with the flow direction perpendicular to the boundary. The water level, composed of thirteen tidal constituents ($M_2$, $S_2$, $N_2$, $K_2$, $K_1$, $P_1$, $Q_1$, $O_1$, $M_4$, $MS_4$, $MN_4$, $P_1$, $MM$, and $MF$), were extracted from the TPXO8 global tide model at the two end points of the open boundary (Egbert and Erofeeva, 2002). The input values at the grid points along the boundary is interpolated between the two end points by MIKE 21.*

*Considering the resolution of TPXO8, i.e. 1/6 degrees (https://www.tpxo.net/global/tpxo8-atlas), we also checked using eight extraction points along the open boundary but the change of the results were unnoticeable, in comparison of using the two end points. This can be attributed to the short length of the open boundary, i.e. 1.88 degrees, in comparison to TPXO8 resolution, and deep water depth at open boundary.*

*To accommodate this comment, the following explanations will be added to the revised manuscript as follows: (page 12, line 170):*

Following Ranji et al. (2016), TPXO global tide model is selected to define the water level at the line open boundary (Egbert and Erofeeva, 2002) with the flow direction perpendicular to the boundary. Thirteen tidal constituents ($M_2$, $S_2$, $N_2$, $K_2$, $K_1$, $P_1$, $Q_1$, $O_1$, $M_4$, $MS_4$, $MN_4$, $P_1$, $MM$, and $MF$) are used to simulate the tide elevations by TPXO8. Constituents are extracted for the two end points of the open boundary and the input values at the grid points along the boundary are interpolated between these end points.

3.The first seven figures show different types of maps of the Persian Gulf. This should be reduced to 1 or 2. Some figures show scientific information (such as Figure 4) without any explanation of how this was derived or who derived it. In total there are 28 figures. This is way too many figures. Future submission should reduce the total number of figures to 10-14.

*Thank you for the comment. Figure 4 was generated out of the field measurements by the authors (please kindly refer to pages 6-7, lines 110-120). For better clarification, its definition will be moved to the previous paragraph in the modified manuscript.*

*We combined figures 2 and 3 into Fig. R2 but it might be better to keep the original figures for the clear visualization of all stations. To accommodate this comment, Figures 5, 6, and 7 were combined into Fig. R3 and figures 12 to 15, 16-17, 21-22, 23-24, and 27-28 were merged together, i.e. Figs. R4 to R8. Thus, the number of figures will be reduced from 28 to 18.*

[Figure]

*Fig. R2: Water level measurements and harmonic constituent data around the PG.*

[Figure]

*Fig. R3: Bathymetry and computational grid (top and middle), two alternative locations of open boundary of the model (middle), and bathymetry data sources from ETOPO and NCC (bottom).*

[Figure]

*Fig. R4: Simulated co-tidal and co-range charts of principal semidiurnal and diurnal constituents. Solid lines denote co-amplitude lines and dash lines denote co-tidal lines.*

[Figure]

*Fig. R5: Map of shallow-water constituents (left) and classification of tides (right) in the PG.*

[Figure]

*Fig. R6: The locations of the nodal lines and nodes for constituents with the Coriolis force (diurnal: bottom right, semidiurnal: top right) and excluding the Coriolis force (diurnal: bottom left, semidiurnal: top left).*

[Figure]

*Fig. R7: Co-tidal charts of principal semidiurnal and diurnal constituents, derived from model results in two scenarios including the constant water depth of 36 m and real bathymetry of PG. The background colors show amplitude differences in the two tests.*

[Figure]

*Fig. R8: Co-tidal charts of principal semidiurnal and diurnal constituents with constant water depth of 36 m in two scenarios with and without bed friction. The background colors show the differences of tidal wave amplitudes in the two cases.*

4. Despite the large number of tidal constituents used and given a lack of comparisons with other models, I am not convinced that this model performs better than previously used models.

*As the field data (times and/or locations) is not the same as employed measurements by other researches, it is not possible to directly compare the performance of present model and past published modeling studies. However, the present modeling might be advantageous to previous ones because of the large employed data set for the verification, i.e. water levels, current speeds and tidal constituents at different locations of PG. The high-resolution bathymetry data on the north coast of the PG (figure 5) is another favorable component of present model, which has only been used in Ranji and Soltanpour (2021).*

*As it was discussed above, the present model has similarities with the numerical model of Ranji and Soltanpour (2021). Figure R9 shows the comparisons between the present model and the*

*model of Ranji and Soltanpour (2021), before the optimization, in three common stations. It is observed that the accuracy both models is almost equal. Using spatially varying Manning coefficient, their optimization process resulted in higher correlations in some stations and more discrepancies in some other with an overall better performance of about +3%, compared to applying a constant friction (please refer to Fig. R1). It should be also added that wind force has also been included in the driving forces of the model of Ranji and Soltanpour (2021). The poor performance of both models at Bahrain (Mina Salman) can be related to the low accuracies of employed bathymetry data, i.e. ETOPO2v2 in our study and GEBCO in Ranji and Soltanpour (2021), in shallow areas of south PG.*

[Figure]

*Fig. R9: Comparisons of water levels at selected stations. Red and black lines respectively present the field data and models (Left: Ranji and Soltanpour, 2021, Right: present study).*

5.The tidal prediction for Kuwait is poor. Why? Do previous models produce similarly bad results?

*Ranji and Soltanpour (2021) also mentioned the poor results in shallow water stations, which got worse after the bed roughness optimization, e.g. the objective functions changed to -2.9 in Kuwait station (Fig. R1).*

*Sabbagh-Yazdi et al. (2007) presented the computed water surface levels and the results of the British Admiralty tide table at Bushehr, Ajman (Middle) and Ras-Al-Khafij. They did not demonstrate the comparisons of model outputs and field data at Kuwait.*

*Pous et al. (2012) also did not compare their modeling outputs at Kuwait. They showed the comparisons of their numerical results of the amplitudes and phases of constituents with those derived from the observations, through the harmonic analysis of International Hydrographic Office data. However, this data source is different from the employed constituents of present study, derived from the admiralty method (Glen, 2015) and admiralty's observational constituents (UKHO, 2005).*

*To accommodate this comment, the following sentence will be added to the revised manuscript (page 13, line 182).*

The poor performance at Kuwait station can be related to lack of high resolution bathymetry data in that shallow water area.

6.   You missed a possibly important reference => Pous et al. (2012) A Process Study of the Tidal Circulation in the Persian Gulf, Open Journal of Marine Science, 2, 131-140, http://dx.doi.org/10.4236/ojms.2012.24016.

*Thank you for pointing out this mistake. We had carefully checked this important study and referred to it in the introduction (page 2, lines 56-58). The error of referring to the other publication of the authors, i.e. Pous et al. (2013) will be corrected in the revised submission:*

Pous et al. (2012) applied a 2D shallow-water model over the northwestern Indian Ocean, forced by seven tidal components at the southern boundary, to derive the co-tidal/co-range charts of harmonic constituents of the PG. They also presented velocities of tidal currents, residual tidal currents, and form factor over the PG.

Pous, S., Carton, X. and Lazure, P.: A process study of the tidal circulation in the Persian Gulf, Open J. Mar. Sci., 2(04), 131–140, 2012.